

# How to reliably detect molecular clusters and nucleation mode particles with Neutral cluster and Air Ion Spectrometer (NAIS)

Hanna E. Manninen[1], Sander Mirme[2,3], Aadu Mirme[2,3], Tuukka Petäjä[1] and Markku Kulmala[1]

[1]Department of Physical Sciences, P.O. Box 64, FI-00014 University of Helsinki, Finland
[2]Institute of Physics, Laboratory of Environmental Physics, University of Tartu, Ülikooli 18, EE-50090 Tartu, Estonia
[3]Airel, Ltd., Observatooriumi 5, EE-61602 Tõravere, Estonia

*Correspondence to*: H. E. Manninen (hanna.manninen@helsinki.fi)

**Abstract.** To understand the very first steps of atmospheric particle formation and growth processes, information on the size where the atmospheric nucleation and cluster activation occurs is crucially needed. The current understanding of the
dynamics of charged and neutral clusters and particles is based on their theoretical prediction and experimental observation. This paper will give a standard operation procedure (SOP) for Neutral cluster and Air Ion Spectrometer (NAIS) measurements and data processing. With the NAIS measurements, we have contributed to improve the scientific understanding by: 1) direct detection of freshly-formed atmospheric clusters and particles, 2) linking experimental observations and theoretical framework to understand the formation and growth mechanisms of aerosol particles, and 3)
parameterizing formation and growth mechanisms for atmospheric models. The SOP provides tools to harmonize the world-wide measurements of small clusters and nucleation mode particles and to verify consistent results measured by the NAIS users. The work is based on discussions between the NAIS users and the NAIS manufacturer.

**Keywords:** Atmospheric aerosols, aerosol instrumentation, ion spectrometer, charged and neutral clusters.

## 1 Introduction

The detailed formation mechanisms and the chemical composition of vapors, which participate in the particle formation processes, have remained unclear until very recently due to lack of direct atmospheric measurements (Manninen et al., 2010; Kulmala et al., 2013, 2014; Ehn et al., 2014). Aerosol particles have global effect on Earth's climate and regional effect on air quality. In atmospheric particle formation, we study the nucleation of stable liquid or solid phase clusters from gas phase precursors. Atmospheric nucleation can happen via molecular clustering, and it is followed by cluster activation for
enhanced growth (Kulmala et al., 2013). Freshly-formed particles grow when different vapors condense onto the particle surfaces or the particles collide and stick together. When aerosol particles grow further to sizes where they can act as cloud condensation nuclei, they start to have effect on climate. In atmospheric aerosol science, one of the main objective is to contribute to the reduction of scientific uncertainties concerning global climate change issues, particularly those related to aerosol-cloud interactions (IPCC, 2013).



Although the Neutral cluster and Air Ion Spectrometer (NAIS, Mirme and Mirme, 2013) is a recently developed instrument, it has been already widely used in atmospheric particle formation studies. First field observations by Kulmala et al. (2007) showed the capability of the instrument for direct detection of the newly formed particles, and later the long-term observations in field lead into fundamental understanding of the cluster formation and activation (Manninen et al., 2009).

The NAIS has been used in various environments at all continents to study both natural and anthropogenic aerosols. Duration of the measurements vary from short-term campaigns to long-term field studies. For example, NAIS field measurements have been made in the boundary layer (e.g. Manninen et al., 2010), and in the middle troposphere (Laakso et al., 2007; Boulon et al., 2011; Rose et al., 2015) and upper free troposphere (Mirme et al., 2010), in the tropics (Suni et al., 2008, Martin et al., 2010, Siingh et al., 2013), at the mid- and high-latitude (Lihavainen et al., 2007; Manninen et al., 2010)

and polar regions (Virkkula et al., 2007), and in remote, rural and urban areas (Tiitta et al., 2007; Backman et al., 2012; Hirsikko et al., 2007, 2013; Herrmann et al., 2014; Jayaratne et al., 2014). Several laboratory studies have been conducted to investigate connection between small cluster ions and new particle formation (e.g. Duplissy et al., 2010; Kirkby et al., 2011; Ortega et al., 2012; Franchin et al., 2015).

In this paper, we will present one of the key methods to measure the number size distribution of clusters at sub-3 nm and

nanoparticles at sub-25 nm (i.e. nucleation mode particles) (see Kulmala et al., 2012). This standard operation procedure (SOP) is based on scientific and technical discussions between the NAIS and the Air Ion Spectrometer (AIS) users among the ACTRIS (Aerosols, Clouds, and Trace gases Research InfraStructure network) partners as well as the (N)AIS manufacturer Airel Ltd., Estonia. The procedure work is led by the University of Helsinki. Aim is to provide consistent results and unified datasets measured with the NAIS around the world, as the NAIS results may be used for developing

aerosol process parametrization for the atmospheric models, and validating and constraining the global models. Note that these procedures apply to instrumentation which are used currently. As the NAIS's are developed continuously, the maintenance, calibration and data processing methods need to be updated and modified as well.

**Special considerations when measuring cluster and nucleation mode particles with the NAIS.** The instrument is capable of measuring the number size distribution of atmospheric ions and particles by collecting signal simultaneously with many

electrometers. The complete distribution of both polarities is measured rapidly with parallel columns. This is the main advantage of the NAIS, but it also creates convoluted instrument construction, and complex maintenance and calibration procedures.

The sampling and detection of small ions and freshly-formed particles is demanding. Firstly, the charging probability of neutral nanometer-sized particles is very low and the concentration of growing freshly-formed particles is often less than ten

particles per cm-3. Thus, corona discharging and large sample-flow rates are essential to increase the amount of collected sample. Secondly, the clusters and small particles can undergo rapid transformations and their composition can change during sampling before the actual detection. Thus, the residence time of the air within the instrument should be short and the temperature, humidity and trace gas composition of the sheath air should be similar to ambient. Thirdly, reducing diffusional





and electrical inlet losses of charged clusters and nanoparticles is a crucial requirement for the measurement set-up. Thus, the inlet lines should be as short as possible, and the inlet must be grounded and there should be no ungrounded conductive or any dielectric material near the inlet. Last (but not least), the calibration and verification of the NAIS under laboratory conditions is essential to confirm that the field results are comparable, and to avoid misinterpretation of data and incorrectly

calculated nucleation parameters.

**Procedure overview.** The SOP (Sect. 3) is written for the NAIS users with different background. The procedure has three main parts:

- *Sections 3.1-3.2* to calibrate and verify the instrument both in laboratory and field. This is essential during a long-term operation, and prior and after short-term campaigns to confirm that the results are reliable.

- *Sections 3.3-3.5* to install, operate and maintain the instrument during a field, laboratory or chamber measurement. Various environmental conditions are considered.

- *Sections 3.6-3.7* to process the collected data including the data corrections and data quality checks. This step is typically followed by the new particle formation data analysis which is described in detail in Kulmala et al. (2012) in a Nature protocols article.

Critical topics are highlighted. In Sect. 4, the troubleshooting is finger-pointing the most typical issues during the NAIS operation. In Sect. 5, the most anticipated ion and particle number size distributions measured with the NAIS are presented.

## 2 Instrumentation

### Neutral cluster and Air Ion Spectrometer

The NAIS is a multichannel aerosol mobility spectrometer capable of measuring the mobility distribution of charged

particles and ions of both polarities in an electrical mobility range from 3.2 to 0.0013 $cm^2V^{-1}s^{-1}$, and the size distribution of total particles in size range from 2.0 to 42 nm. A controlled charging of the aerosol sample with a needle-corona charger, together with electrical filtering of the corona-generated ions, is used for measuring the total aerosol particles. Manninen (2011) and Mirme and Mirme (2013) introduced the principles of NAIS design and raw signal processing in more details.

**Instrument schematics and flow chart.** Figure 1 illustrates that the NAIS can have a 1-blower, 3-blower or 4-blower flow

system. In the 1-blower system all flows are controlled by one blower. In the 3-blower system there is one sample flow blower for the instrument, whereas in the 4-blower version both columns have a separate sample flow blower. All the flowrates – sheath flow for both polarities and sample flow (either single or separate) – are measured using Venturi tubes and differential pressure sensors. The blowers are automatically controlled to maintain the correct flowrates. The flow sensors are calibrated at Airel Ltd., Estonia.



**Measurement modes.** The mobility analyzers of the NAIS are preceded by a software controlled sample preconditioning unit. Depending on the measurement mode of the instrument, the unit may filter particles to measure the zero signal, charge the particles to measure neutral aerosol, or leave the sample untouched to measure naturally charged ions.

The four main measurement modes of the instrument are: 1) *Ions mode* is used for measuring naturally charged particles, i.e. ions. All parts of the preconditioning unit are switched off, so the aerosol sample is not modified. 2) *Particles mode* is used for measuring all particles including the uncharged fraction. The main charger and post-filter are switched on. 3) *Alternating charging mode* is similar to the particle mode but additionally the discharger is switched on. This has the effect of "neutralizing" the sample and so it improves the instrument performance in case of non-steady-state charge distribution. 4) *Offset mode* is used for measuring the zero signal and noise levels of the electrometers. The particles are first charged by the discharger with ions of the opposite polarity to what measured by the analyser and then filtered. This way no detectable particles will enter the analyser.

**Recommended measurement cycle** for the NAIS is [offset 30, ions 90, particles 90]. Reliable offset measurements are vital for the accuracy of the instrument itself and, clearly, the final number size distribution. The offset signal is estimated using a linear regression on the electric current measurements from previous and next offset measurement cycles. The variance of the electric current signal during the offset cycle is used to estimate the noise level of individual electrometers. It is recommended that the duration of the offset measurement is between 30 and 60 seconds, and the total length of the measurement cycle is 2 to 5 minutes.

**Sample preconditioning unit with corona chargers and electrical filters** is represented in Fig. 2. Although bipolar radioactive chargers are the most widely used chargers due to their well-defined charge distribution (Wiedensohler, 1988; Reischl et al., 1997), unipolar diffusion chargers can attain much higher charging levels (Intra and Tippayawong, 2009). Thus, the NAIS uses unipolar corona discharge ionization. The high voltage (HV) supplies feed the corona charger needles (typically 2–3 kV). The voltages are controlled by a feedback system to maintain a constant electric current of the corona ions to the outer electrode of the charger volume.

Both measurement columns use two corona chargers. The first charger is called the discharger and it can charge particles in the opposite polarity of what is measured by the succeeding mobility analyser. The discharger currents are –20 nA for the positive column and 20 nA for the negative one during the offset operating mode. During alternating charging mode the currents are –10 nA and 10 nA, respectively. The discharger is followed by an electric filter which is used during the offset operating mode to remove the charger ions and charged particles. Although these particles would not be directly detected by the analyser due to their wrong polarity, the space charge would still induce unwanted electric currents to the measurement electrodes. The voltage of the filter is 500 V, when switched on.

The main charger current is set to 25 nA for the positive charger and to –22 nA for the negative one. The difference in charging currents compensates for the difference in the charging efficiency of the chargers (due to the different mean mobility of positive and negative ions, Manninen et al., 2011).



**Notice when using the particles mode.** The corona charger ions have a diameter range of 1.0–1.6 nm (1.3–0.8 $cm^2V^{-1}s^{-1}$, Manninen et al., 2011). These sizes define the absolute lower detection limit of the NAIS in particle mode. The electrical filtering of the charger ions and inability to remove all the naturally charged particles plays an important role in determining the lowest detection limit, see Fig. 3. As can be seen from Fig. 4, the post-filter is used to cut-off the corona ions generated

by the charger, and consequently the small charged particles are filtered out together with ions used for charging. If the corona ions were allowed to pass into the analyser they would saturate the first measurement channels and cause invalid signal in later channels. The post-filter voltage is typically 30–150 V.

## 3 Procedure

We encourage all NAIS users to follow to this procedure which is based on earlier scientific work by Mirme et al. (2007,

2010), Asmi et al. (2009), Manninen et al. (2009, 2011), Kulmala et al. (2012), Mirme and Mirme (2013), and Wagner et al. (2016). The procedure has been motivated by the need of reliable long-term field measurements and by the comparability of such long-term field data. We aim to improve the comparability of the results by improving the instrument's verification, maintenance, and data processing procedures.

### 3.1 Instrument calibration

Prior to the measurements, the NAIS flow sensors and the electrometer background levels should be checked. The NAIS software monitors the flows using Venturi flow meters continuously during the measurements. The NAIS flows should be compared to a reference flow meter. If a discrepancy is detected the flow sensors will need to be recalibrated to update the calibration coefficients in the measurement software. The background can be checked by passing particle-free air into the instrument or by performing concentration calibration as a function of particle size. The voltages in the inner electrode of the

differential mobility analyser (DMA) should be measured before and after ambient measurements. Also the background and DMA voltages are continuously recorded by software during the measurements.

It is recommended that ion spectrometers take part in the calibration and intercomparison workshops organized in co-operation by University of Helsinki and Airel Ltd. The goal is to organize these workshops on regular basis. During the workshop the ion spectrometers are mobility and concentration verified and flow calibrated.

### 3.1.1 CRITICAL: Determining the flows of the NAIS

The sheath and the input flows of the (N)AIS are critical for the precise determination of the particle mobility and concentration. Thus, prior to the mobility and concentration calibrations the instrument should be cleaned, leak tested and flow checked. The cleaning procedures are essential before the determination of flows. If dirt is deposited onto the tubing or nets inside the instrument, the flow resistance will alter the volume flow through the Venturi tubes (see Sect. 3.5.2). When

maintenance cleaning is done regularly, the instruments can perform well for long periods and flows stay stable (Gagne et





al., 2011). This is primarily relevant for instruments with the 1-blower flow system as the correct flow balance is very delicate. The newer instruments with 3- and 4-blowers will maintain flow stability for much longer periods without a maintenance and will only require recalibration in case the simple check indicates a problem.

**Leak tests** can be done using several methods. *Alternative 1:* The volume flow rate measured from the inlet and outlet
should match when no leaks exist. This is sufficient in case the instrument is operated at atmospheric pressure and a low pressure drop inlet is used. *Alternative 2:* By blocking the inlet and outlet, applying overpressure or underpressure inside the instrument, and measuring the flowrate required to maintain the constant pressure. The recommended range is 50–100 mbar over- or underpressure. Overpressure above 200 mbar may damage the instrument. *Alternative 3:* When the inlet is closed, the recorded sample flow rate should drop to well below 10 LPM. This method is suitable for a quick check however it is not
completely reliable.

The flow calibration should be done once in two years during a long-term operation, or before and after a short-term campaign measurement. However, the flows should be determined always when a blower is replaced or a large leak sealed.

**Option A: Flow calibration with pressure sensors for 1-blower systems**

For a 1-blower system, in Fig. 1a, one blower runs all the flows. Five flow rates (sheath and output flows of both analysers
and the total exhaust flow) are measured with the Venturi tubes. In normal operation, only the exhaust flow rate is measured continuously, as it is the most sensitive to the changes of all flows. Each Venturi tube has an individual calibration where the exact pressure drop, corresponding to the specified flow rate, is determined in NTP-conditions (Venturi calibration values provided by Airel Ltd.). This procedure is illustrated in Fig. 5. A Venturi calibration is not needed done by the user, but the pressure drops should checked and adjusted if needed. The differential pressure over the 5 Venturi's should be measured
with a reference differential pressure measuring instrument (e.g. TESTO 512 0–2 hPa), and checked against the value obtained from the calibration to be sure that the flow rate though the Venturi is correct. If the checked pressure differences do not correspond to the calibrated ones, they should be adjusted to match while keeping a constant overpressure (80 mmH$_2$O) after the blower (measured with e.g. TESTO 512 0–20 hPa).

Determining the flows for 1-blower system: First, the user verifies that the sample and outlet flows are equal that no leaks
exist. Second, the blower power should be adjusted so that the overpressure after the blower is 80 mmH$_2$O. Third, each of the five flows are measured via the pressure difference from the Venturi tubes, and adjusted if required to match the values given in the Venturi calibration (while keeping a constant overpressure after the blower). Last, the user should check that the sampling flow is kept at ~60 lpm. For a full pressure sensor calibration the instrument user will require assistance from Airel Ltd. to provide detailed information on the flow calibration set-up and update the measurement software.



**Option B: Flow calibration with flowmeters for 3- and 4-blower systems**

For the 3- and 4-blower system, in Fig. 1b-c, the calibration (actually a flow verification) involves measuring the volumetric flow rates with an external flowmeter (e.g. TSI 4000 series) at the instrument's exhausts and after the sheath air filters. First, place a reference flowmeter at both exhausts (measure one exhaust flow tube at time) to verify the sample flow Venturis

which are located just before the exhaust tubes. The sample flow and exhaust flow are assumed to be the same when the instrument is not leaking. Second, disconnect a sheath flow tube and place the flowmeter after the sheath air filter to verify the sheath flow Venturis which are located prior to the blower and filter. Figure 6 shows a quick and easy way to check the sheath flow of the negative polarity for the 4-blower system. Repeat this for the both polarities sheath flow. For a simple instrument check it is sufficient to compare the reference flowmeter value to the corresponding NAIS flowmeter value. Note

that the NAIS flowmeters show the actual volume flow rate that is not adjusted to the standard conditions. A full flow sensor calibration can be done by the instrument user but it will require assistance from Airel Ltd. to provide customised calibration software, process the results and update the measurement software.

In case of the 3 and 4-blower systems, the instrument also includes a barometric pressure sensor that is actively used for determining the correct flow rate. The instrument sensor value should be compared to a reference barometric pressure sensor

and the calibration coefficients adjusted if necessary.

In case the 3-blower system, there is only one sample flow Venturi sensor in the instrument that measures the total flow from both analysers. An additional step is required for confirming the sample flow balance of the negative and positive analyser. This involves adjusting the two valves before the Y-connector where the sample flows join from both analysers. The two Venturis next to the valves should be measured simultaneously using two handheld differential pressure sensors. The

calibrated values for these pressure differences are either written inside the instrument or available from Airel Ltd. The valves should be adjusted accordingly.

**3.1.2 CRITICAL: Determining the voltages of the NAIS**

The response of the DMA high voltage (HV) supply should be followed from the instrument diagnostics. Correct sizing of small ions and particles in the DMA is highly sensitive to accurate knowledge of the applied HV. Particular care is, hence,

required in the low voltage range. The voltage in the inner electrode are ±9, ±25, ±220, ±800 V, depending on the polarity of the DMA. On the other hand, for the NAIS with serial number larger than 29 the corresponding voltages are ±9, ±35, ±150, ±700 V. For confirming the voltages with an independent measurement, a HV-probe with ultralow impedance should be used.

**3.1.3 Determining the losses and sizing accuracy of the NAIS**

In the size range of the cluster ions and small neutral particles the calibration is a challenging task due to capability of reference instruments, proper calibration aerosols, and instrumentation for the size-separation. Possible calibration set-ups



are presented in detail by Asmi et al. (2009), Gagné et al. (2011), and Wagner et al. (2016). The NAIS should be calibrated using an extensive set of instrumentation: 1) in the sub-3 nm size range the high resolution H-DMA (Herrmann-DMA; Herrmann et al. 2000) is used for size separation to determine the transfer function at 1–10 nm, 2) monomobile standards (Ude and De la Mora 2005) are used for a mobility calibration, 3) a Hauke DMA (Winklmayr et al. 1991) is used for

mobility and concentration calibration in the size range of 4–40 nm, and 4) the flows are determined as descripted above. Wagner et al. (2016) studied the accuracy of the NAIS in a supplementary laboratory calibration. They concluded that in ion mode the sizing of the NAIS was very accurate, regardless of the version of the data inversion, and the ion number concentrations were underestimated 15–30%. Using a correction introduced by Wagner et al. (2016), the uncertainty of the ion concentration measurement of the NAIS can be reduced to ~10%, allowing the NAIS to be used in quantitative ion

cluster and charged particle studies.

### 3.2 Instrument verification

Prior to measurements, in addition to the electrometer background levels, the balance of number concentration measured with the positive and negative columns should be checked. Between calibration intervals, if possible the NAIS should be regularly compared to a reference instrument for a period of few days per year, especially during a long-term operation.

Verification should be done always after instrument transportation.

#### 3.2.1 Intercomparison to other instrumentation

It is recommended that different methods would be used in parallel to determine the cluster and nucleation mode number concentration in order to avoid artefacts (Kulmala et al., 2012). The intercomparison is recommended to be done within an intercomparison workshop or at sampling site if a reference instrumentation is available (e.g. ion spectrometer, gerdien

counter, condensation particle counter, differential/scanning mobility particle sizer, or air conductivity measurement, e.g. Asmi et al., 2009; Gagné et al., 2011).

#### 3.2.2 CRITICAL: Balance between negative and positive measurement columns

To verify instrument operation prior to measurements, the balance of number concentration measured with the positive and negative columns should be checked. The concentration verification can be done by generating population of sample aerosol

(in equilibrium charge distribution) and measuring with both columns. Good agreement (with 10–20%) between the polarities gives confidence that the instrument is working properly.

#### Option A: Orange peeling experiment

Peeling citrus fruit and releasing D-limonene, which is a common monoterpene, into the room air can lead to new aerosol particle formation in indoor environment (Vartiainen et al., 2006). Thus, this is fast, easy and cheap way to generate

nucleation mode aerosol particles over the whole size range of the NAIS when the D-limonene oxidation products triggers



the particle formation and subsequent growth (Gagné et al., 2011). Finding the right amount of fruits to peel to generate enough but not too much vapours can take few trials. Note that there should be sufficient ozone concentration and low background aerosol concentration in the room.

**Option B: Indoor and outdoor sampling experiment**

Fast concentration verification between the polarities can be done by sampling from indoor and outdoor. Indoor sample due to efficient air-conditioning is typically dominated by cluster ions and can be used to check the balance between small ions, whereas outdoor sample has an abundant Aitken mode and can be used to test larger ions and particles (Hirsikko et al., 2007).

**Option C: Test with small ions from external charger**

Extensive number concentration range ($\sim 10^2$–$10^6$ cm$^{-3}$) of both positive and negative small ions can be easily and quickly generated by using an external radioactive source (Steiner et al., 2014). This bipolar charger should be placed right in front of the NAIS inlet. By varying the flow rate through the external charger, the user can vary the number concentration of the charger-generated ions.

**3.3 Instrument installation**

**3.3.1 CRITICAL: Inlet design and installation**

Recommended inlet design is a 50 cm long metallic tube (of diameter 35 mm) with a bend (90° angle facing down) and metallic net (grid size 1 mm) in the end of the inlet line, see Fig. 7. The brass inlet connector with a metallic net inside (sold by manufacturer, Airel Ltd.) is recommended to be used. The instrument is operational both in vertical and horizontal position. However, the vertical orientation for inlet line is not recommended. Recommended inlet height is 2 meters above

the ground level. Note that the electrode effect can cause imbalance between polarities (negative and positive ion number concentrations) when inlet height is below 4 meters from the ground level. The ionosphere has positive charge and Earth's surface has a negative charge. Thus, Earth's surface works as a negative electrode which attracts positive ions and rebels negative ions close at ground level (Hoppel, 1967).

Inlet lines should also include a proper rain cover as the rain droplet can interfere the measured spectra (Tammet et al.,

2009). In field conditions, make sure that rain does not get into the instrument because of the high sampling flow rate. If there is a chance of water dripping on the inlet then the end of the inlet tube should point at least slightly downwards. The mobility analyser is sensitive to insects and fibres which may deposit on the electrodes or electrometers and cause corona discharge, noise and parasitic currents. Furthermore, the pressure drop from the inlet to the instruments should be kept in the range of few hPa.



**Option A: Inlet with minimized diffusional and electrical losses**

Due to diffusional losses of small particles the inlet lines needs to be kept as short as possible and as straight as possible, and the flow should be kept close to laminar. Enhanced diffusional particle losses may occur in sampling lines containing bends or elbows. These enhanced particle losses increase with a decreasing radius of the bend or elbow. It is very essential that the

inlet lines and connectors should be made from conductive material (preferably stainless steel) to avoid losses caused by static electric charge. Experience has shown that non-conductive tubing (e.g. plastics) may remove a considerable fraction of any charged particles by electrostatic forces. A rough estimation for particle losses should be done on the measurement site after the installation by measuring with and without inlet set-up.

**Option B: Inlet with aerosol sample conditioning**

When working in a warm and moist atmospheric environment, the dew point temperature can reach the standard temperature of a measurement cabin or container (20–25°C). This requires that the aerosol sample flow has to be dried, either directly in the sampling line or at the instrument. E.g. in rainforest a *drying by heating* has been used for the NAIS. The inlet line and at the instrument is heated to avoid water condensing into the tubing (sample line and instrument always at higher temperature then ambient). To limit RH in the aerosol sample flow, we don't recommend using a membrane dryer (e.g. Nafion™ dryer),

or a silica-based aerosol diffusion dryer due to high aerosol sample flow rates and increased diffusional losses of clusters and small particles. When sampling in highly polluted environment, we recommend to add a core sampling inlet if absolutely needed to dilute sample with aerosol-free bypass flow (*drying by dilution*, see Wagner et al., 2016).

### 3.3.2 Sampling location and measurement set-up

The NAIS should be placed indoors in an air-conditioned scape. The instrument should be operated in an environment

temperature of 5–35°C to avoid a malfunction of the electric circuits (chargers and filters, HV supplies). The close range of the instrument inlet should be grounded well that sample ions are not attracted by the static charge on nearby surfaces. The instrument itself is grounded via the power cable. Measurement cycles should be set based on scientific aims.

**Option A: Ground site measurement**

In close to sea level (NTP conditions), the NAIS software uses a fixed sheath-flow rate value of 60 LPM and a sample-flow

rate value of 30 LPM (1-blower system) and 27 LPM (3- and 4-blower system) per DMA when calculating the ion and particle number size distributions. The deviation of the flow rate from the default value should be then taken into account during data processing by applying a correction to the number size distributions. The volumetric flow rates are typically recorded by the instrument, however the measured distributions are not automatically corrected in case the flow rates deviate from the specified values. For the ground-based measurements recommended measurement cycle is [offset 30, ions 90,

particles 90].



**Option B: High-altitude site measurement**

At high-altitude sites, the NAIS (with 3- and 4 blower systems) volumetric sample flow rate is kept constant whereas the sheath-flow rate is varied automatically (Mirme et al., 2010). Automatic adjustment of the sheath-flow compensates changes in the particle mobility in exceptionally low or varying air pressures and temperatures. Thus, the classified particle size range

is kept invariant of air pressure and temperature changes. The effect of temperature variations is considered small because of warming in the sampling line. In case of 1 blower system, we recommend upgrading into 3- and 4-blower system. Otherwise, the user needs to manually keep a blower operating in the right volumetric flow range, and separate airflow calibration is needed for the 1-blower system.

**Option C: Flight measurement**

Several improvements were made to the airborne NAIS to able to measure ambient air at varying altitudes while being situated inside a pressurized aircraft (see Mirme et al., 2010). When measuring inside a pressurized aircraft, the instrument leaks have to be controlled well. We recommend to use the upgraded 3- and 4-blower system at flight measurements. These NAIS's automatically adjust its aerosol sample- and sheath-flow rates so that the particle sizing and volume sample-flow rate remains constant regardless of air pressure. Typical modifications to airborne measurements are following: 1) replacing the

electricity supply to match the system used in aircraft, 2) designing a special inlet tube system to sample air from outside the aircraft, 3) reinforcing instrument rack and attaching it into aircraft frame, 4) modifying the instrument for increased air tightness in case it is used in a pressurized cabin, and 5) setting the length of the measurement cycle to minimum. If the inlet has an over pressure, the exhaust flow may need to be restricted with a valves (e.g. adding some soft tubing and a pinch cock). For airborne measurements recommended measurement cycle is [offset 30, ions 45, particles 45].

**Option D: Laboratory and chamber measurement**

In aerosol chamber measurements, where it is required to minimize the amount of air withdrawn, the high sample flow rate of the NAIS is a challenge. The NAIS is recommended to operate with a recirculation system, which dilutes the inlet sample flow with filtered air coming from the exhaust of the instrument. In other words, sample air from the chamber is diluted with a portion of the exhaust air of the instrument, which is filtered with a High Efficiency Particulate Air (HEPA, e.g. Camfil

Megalam, MD 14-305X305X66-10) filter and mixed with the sample air (in more details, see Franchin et al., 2015). The pressure drop on the filter and dilution mixer should be below 10 mbar to ensure that the sample flow blowers of the NAIS are able to comfortably circulate the air. The use of the dilution system allows reduction of the withdrawn flow from 54 to 20–30 LPM. Otherwise in laboratory measurements, where the available sample flow rate is limited and sampled concentrations are small, it is recommended to use only one polarity (column) of the NAIS (e.g. Manninen, 2011) at a time

depending on the polarity of the generated aerosol sample. This is possible only for the 4-blower system where the columns work completely independently. To disable one column, disconnect and close the corresponding tube at the inlet Y-





connector which divides the flows to the two analysers or alternatively close the corresponding exhaust outlet. The instrument configuration files should be modified to switch off the blowers for the disabled column. This must be done when the exhaust outlet is closed to avoid damage to the blowers. When using NAIS in exposure studies with extremely high concentrations ($\sim 10^5$–$10^6$ cm$^{-3}$), we recommend to add core sampling inlet.

**3.4 Monitoring and adjusting instrument parameters in varying environmental conditions**

A large number of measurement parameters are automatically monitored by the instrument. This includes for example flow rates, blower control signals, charger currents and filter voltages. Most of the parameters are continuously checked by the Spectops measurement software and diagnostic warning flags are indicated if a problem is detected. The presence of warnings does not definitely mean that the measurements are invalid. The user should always understand and confirm the

reason why a warning is given and fix the issue when necessary. The absence of warnings does not guarantee that the measurements are correct.

The corona currents and filter voltages are adjusted by varying the HV supply feed voltage. The discharger and the main charger are controlled with a feedback loop driven by the current measured from the surrounding electrode. The post-filter is controlled according to the current measured on the first electrometer channels in particles measurement mode. The blowers

are also actively controlled according the measured flow signals from the Venturi tubes.

The automatic adjustment with feedback works as long as all feedback controls are between 0.1 and 4.9 V which the NAIS user should monitor. A sensor value starts to deviate from a target value if the control voltage goes too close to 0.0 or 5.0 V. For example, in Spectops diagnostic a *sensor value* is the value measured by the analog-digital converter. It is converted to the actual parameter value by some equation (e.g. "airflow sensor" to "airflow"). Parameters which are automatically

controlled by digital feedback have a control and a target value. *Target value* is the ideal sensor value, the goal (e.g. for sample airflow speed the target value equals the sensor voltage that would match 54 l/min). *Control value* is the output voltage of the digital-analog converter that adjusts some function in the instrument (e.g. airflow blowers). The algorithms will automatically adjust the control value so that the sensor value matches target value (e.g. if airflow sensor is below airflow target the airflow control is increased).

**3.4.1 Monitoring instrument performance**

Instrument operation should be checked daily by the user. Figure 8 summarizes a recommended checklist for instrument's performance monitoring. Airel Ltd. provides tools for this. There are two programs in the measurement software package provided with the NAIS: 1) *Spectops* for running the measurements and viewing online data, and 2) *Retrospect* for viewing and reprocessing the recorded data offline. The Airel Ltd. has an extensive Diagnostic checklist in the NAIS manual:

http://wiki.airel.ee/Docs/NaisManual. We will not be repeated here but we recommend the user to follow the checklist.





### 3.4.2 Flow measurement control settings

The blowers are automatically adjusted using a software digital feedback loop controller based on the flow sensors and barometric sensor so that the mobility analysis and sampling volume flowrate remain constant. Thus, when measuring using an inlet with a high pressure drop or in polluted conditions, the blower might be under heavy strain when the flow resistance caused by the air pollution (instrument getting dirty) is overcome by operating the blower with higher power.

### 3.4.3 CRITICAL: Controlling corona discharging

To keep the corona charger efficiency at a constant level independent of environmental conditions, the corona needle voltage is adjusted by varying the HV supply feed voltage according to an active feedback loop. The corona charger's efficiency is directly determined by the charger ion concentration, which is proportional to the electric current of corona ions to the surrounding electrode of the charger volume. Thus, the discharger and the main charger are controlled with a feedback loop driven by the current measured from the surrounding electrode. In normal operating conditions of the NAIS, the corona-needle voltage is in the range of 2–3 kV. Over time the electrode (i.e. corona needle/wire) gets worn out or dull because of dirt depositing on the needle. This will typically cause the corona voltage fluctuate as the corona ignition voltage increases and a stable discharge can no longer be maintained. For this reason the NAIS corona needles need to be cleaned regularly.

### 3.4.4 CRITICAL: Adjusting the post-filter (i.e. electrical filter in particle mode)

In particle mode, the post-filtering settings should be optimized according to environmental conditions. Typically, the filter operates with a voltage of 40–100 V. The corona-generated ions are at the same size range as the smallest particles measured by the NAIS. The set point of the post-filtering voltage is a compromise between the removal of corona-generated ions and the penetration of small aerosol particles. The main electrical filter voltages are adjusted by varying the HV supply feed voltage according to an active feedback loop. In the 3- and 4-blower systems, the user can change the target values by modifying the configurations when using the automatic adjustment by Spectops software (Mirme and Mirme, 2013). In the 1-blower systems, the post-filtering voltage is adjusted by the user manually. Figure 9 shows standards when to change settings or adjust manually the post-filter during the particle measurements. We recommend that in continuous field measurements the automatic adjustment is used (if possible), whereas in laboratory experiments with rapidly changing or unusual aerosol distributions, the automatic adjustment should be switched off (Manninen et al., 2012).

### 3.5 CRITICAL: Maintenance requirements

As important, as the instrument verification, is a regular maintenance of the NAIS to maintain the calibration during the operation. Maintenance procedures include instrument cleaning, leak testing, and checking condition of corona-needles, proper isolation between inner and outer electrode, and proper instrument grounding and inlet operation.





### 3.5.1 Inlet cleaning

The inlet net and inlet tubing should be cleaned thoroughly with 1–3 week interval to maintain the optimal aerosol sample flow and reduce the amount of dirt depositing on the analyzer.

### 3.5.2 Routine instrument cleaning

During long-term operation, the NAIS's should be cleaned at 1–3 month intervals due to the deposition of particulate matter inside the instruments. Within the cleaning procedures all parts which are in contact with the sample- and sheath-flow should be thoroughly wiped with alcohol (e.g. 2-propanol) using delicate task disposable wipes (e.g. Kimberly-Clark Kimtech Science Kimwipes). Wipes which get easily worn out and leave fibres should be avoided. The metallic nets inside the Venturi flow tubes (i.e. tubes with narrow slits for adjusting the volume flow) should be cleaned carefully. When dirt is

deposited onto the nets, the flow resistance to increases and consequently the volume flow through the Venturi tubes decreases and affects the mobility classification. Ultrasonic bath is recommended for cleaning these nets. Instruments with 3 or 4-blowers have typically honeycomb laminarisers inside the Venturi tubes instead of nets. These are less likely to become dirty and careful cleaning with a brush or pressurised air is sufficient. Overall the 3 and 4-blower instruments are significantly less susceptible to flow deviation issues. The corona‐needle chargers should be cleaned (scrape with a sharp

knife) with regularity (1–3 month interval) to make sure that the corona‐generated ion concentration is maintained at a constant level.

**Notice while cleaning the NAIS.** Use plenty of isopropanol and Kimwipes. Do not scratch surfaces of the NAIS. Clean all the surfaces which are in contact with the sample and sheath air flows. Do not wipe the plastic parts with isopropanol, clean them with deionized water to avoid leaving a conductive film on the surface. Always wear gloves when handling the DMAs

and sample preconditioning units. After cleaning procedures, check that the ion and particle number size distributions are continuous before and after cleaning.

### 3.5.3 CRITICAL: Cleaning the analyser's electrometer rings

The number concentration of ions and aerosol particles are determined by measuring a current delivered by a flow of charged particles to an electrometer. The electrometers are extremely sensitive. The deposition of dirt onto the electrometer ring can

deteriorate the signal‐to‐noise ratio of the electrometer. Dust or fibres deposited on the electrode may start coronating in the electric field of the analyser. This is the reason why the electrometers facing the bottom inner electrode, which has the highest voltage, are the most likely to become noisy (electrometers no. 13-21). We recommend to wipe the electrometer ring(s) clean when the average current signal of certain electrometer increases above few tens of fA ($10^{-15}$ A) during the offset measurement mode.





**Option A: By lifting away the sample preconditioning units**

To clean the electrometer rings without removing and opening the mobility analyser, remove the top part of the instrument (i.e. sampling preconditioning unit, in Fig. 1) and use the cleaning rod to clean the mobility analyser by moving it from top to bottom. Figure 10 shows how to do this. Before lifting up the top part of the NAIS, notice to remove all the nuts holding

the plates together and disconnect cables coming from the main compartment of the instrument, and sheath air tubing between the top and bottom part of the NAIS (on both polarities). The top part should be lifted and placed onto a clean surface, while the open analyser should be covered to avoid dropping more dirt into it while cleaning it. Now the inner and outer electrode of the mobility analyser (i.e. electrometer rings) can be cleaned with the cleaning rod by moving it up and down inside the analyser. Avoid scratching the metallic surfaces. The numbering of the electrometers starts from the top to

bottom. Remember that the electrometers detecting the smallest charged particles are on the top of the analyser.

**Option B: By opening the mobility analyser**

To open the mobility analyser it needs to be lifted away from its place. The outer electrode of the analyser should be lifted, and separated from the inner electrode. Supplement S1 shows in details how to clean the analysers by opening the mobility analyser. The electrometers are located on the surface of the outer electrode, whereas the inner electrode has the four voltage

sectors to generate the electric field. Take care not to scratch the inner electrode against the outer electrode. After the electrodes are separated, it's possible to clean the electrometer rings by wiping with a clean cloth and some strong solvent like alcohol or isopropanol, see Fig. 11. Wiping should be done by starting from the centre of the electrode and moving towards the top. Take particular care to clean the gaps between the electrometer rings as well as their surface. Flip the electrode upside down and repeat the operation to clean the bottom electrometers.

**3.5.4 Cleaning or replacing a corona needle**

The charging efficiency of the corona charger can change over time, since the electrode (i.e. corona needle/wire) gets worn out. For this reason the NAIS corona needles need to be cleaned regularly or replaced. To clean the corona needles located in the preconditioning unit's charger or discharger, shown in Fig. 12, or in the sheath air filter, shown in Fig. 13, they need to be opened and removed. As done in Fig. 14, the corona needle can be cleaned from dirt collecting on the tip by gently

scraping the tip with a sharp knife or by trying dissolve the dirt. The corona needles breaks and bends easily so no pressure should be applied. When cleaning of the needle is not enough for efficient charging, the needle needs to be replaced. When replacing a corona needle take care and handle the needle with a flat tip tweezers.

**3.5.5 Replacing a blower**

A blower needs to be replaced when the active feedback cannot maintain the right volumetric flows for sample and sheath

flows which leads to wrong sizing of the aerosol particles. The blower must be sealed properly and the leak testing should be





done for the instrument before determining the instrument flows. For the 1-blower system, the blower sealing should be done using a silicone (e.g. Bostik silicone universal) to seal both the blower itself and to connect it leak-tight to the metal casing, shown in Fig. 15.

### 3.6 Data inversion

#### 3.6.1 CRITICAL: Electrical mobility to mobility diameter conversion

The particle diameter is not a well-defined concept at very small sizes or for highly non-spherical particles and agglomerates. We recommend to use the electrical mobility diameter as it can be converted back to the particle electrical mobility, which is the measured quantity by the NAIS. The electrical mobility ($Z_p$) to particle diameter ($D_p$) conversion should follow the international standard ISO 15900, and use a Millikan-Fuchs equivalent electrical mobility diameter which is based on Stokes

law (Hinds, 1982)

$$Z_p = \frac{neC_c(D_p)}{3\pi\eta D_p},\tag{1}$$

where n is number of excess elementary charges $e$ carried by the particle , $\eta$ is viscosity of air and $C_c$ is the Cunningham slip correction factor for molecule slip taking account of relation between particle radius and mean free path $\lambda$ of the gas molecules (67.3 nm at 296.15 K and 101.325 kPa, Allen and Raabe, 1982).

$$C_c(D_p) = 1 + \frac{2\lambda}{D_p}\left(1.165 + 0.483e^{-0.997\frac{D_p}{2\lambda}}\right)\tag{2}$$

Gas pressure and temperature are included above because they are arguments of viscosity and the Knudsen number ($K_n = 2\lambda/D_p$). The constants used in equations follow the ISO15900 standardization as well Kim et al. (2005):

Air viscosity is

$$\eta = \eta_0\left(\frac{T}{T_0}\right)^{\frac{3}{2}}\left(\frac{T_0+110.4K}{T+110.4K}\right),\tag{3}$$

where $\eta_0 = 1.83245\cdot10^{-5}$ kg m$^{-1}$s$^{-1}$. $T$ is the air temperature and $T_0$ is a reference temperature (296.15 K). On the other hand, mean free path at reference temperature $T_0$ = 296.15 K and reference pressure $p_0$ = 101.325 kPa, when is chosen $\lambda_0$ = 67.3 nm.

$$\lambda = \lambda_0\left(\frac{T}{T_0}\right)^2\left(\frac{p_0}{p}\right)\left(\frac{T_0+110.4K}{T+110.4K}\right)\tag{4}$$

The classic Millikan equation may be inaccurate when studying the finest charged nanometer particles and ions (e.g.
Tammet, 2012, and references therein). Mäkelä et al. (1996) analysed phenomena in detailed and compared different mobility equivalent diameters. For the NAIS data, the mobility to mobility equivalent diameter conversion is commonly



done following suggestion by Mäkelä et al. (1996) using the Cunningham slip correction factor with $\lambda = 64.5$ nm at 296.15 K and 101.325 kPa.

$$C_c(D_p) = 1 + \frac{2\lambda}{D_p}\left(1.246 + 0.420 e^{-0.87\frac{D_p}{2\lambda}}\right) \tag{5}$$

This approach is in excellent agreement with the ISO standard at NTP conditions. Note that the geometric diameter (i.e. Tammet's mass diameter; Tammet, 1995), which is related to particle mass, is about 0.3 nm smaller than the electrical mobility diameter (Mäkelä et al., 1996).

### 3.6.2 CRITICAL: Raw signal: electrometer currents

The raw signal of the NAIS is calculated as follows. The current $I$ (A) measured by one of the NAIS electrometers is related to the sampled aerosol concentration $N$ (ions/cm$^3$)

$$I = N n_p e Q_s, \tag{6}$$

where $n_p$ is the average number of elementary charge units per particle (assumed to be unity), $e = 1.6 \times 10^{-19}$ C is the elementary unit of charge, and $Q_s$ (cm$^3$ s$^{-1}$) is the volumetric sample flow rate passing the electrometer. When the average number of elementary charge units per particle is known, the aerosol concentration can be calculated from Eq. (6).

The NAIS measures all electrometer signals about 12 time per second. Averaged records are calculated typically over 1 second, 10 second and measurement cycle (block) periods. The instrument measures a large number of secondary parameters that describe the detailed state of the whole system. The average electrometer signals together with the secondary measurement parameters are stored into record files by the Spectops measurement software (Airel Ltd., Estonia).

The average electrometer signals are converted into ion mobility or particle size distributions by the Spectops software (Airel Ltd., Estonia) and stored in spectra files. The distribution is calculated using the generalised least squares method to find the size or mobility distribution that best matches the measured electrometer currents according to the instrument matrix while taking into account the noise level estimates. The instrument matrix is based on a mathematical model of the instrument that considers particle losses, charging probability (in case of particle mode measurements), electric field and air flow inside the mobility analyser.

It is important that the user always stores the record files together with the spectra files containing measured distributions. The secondary measurement parameters stored in record files are vital for confirming the validity of the measurements and diagnosing problems with the instrument. The spectra files can be recalculated from the electrometer signals stored in record files.



### 3.6.3 Assumption of equilibrium charge distribution

The sampled particles are assumed to be in charge equilibrium. The particle charging probability is predicted by Fuchs' diffusion charging theory (Fuchs and Sutugin, 1971). At a constant corona-wire current the aerosol charging depends mainly on the particle size, charger ion concentration and the residence time of the aerosol in the charging region. The product of the latter two is called *nt*-product. The model that calculates the NAIS inversion takes into account aerosol volume flows, particle charging probabilities, the loss factor, and the charging parameter (i.e. *nt*-product). The charging probabilities use a calibrated charging parameter $\alpha = 6$, which translates to $nt = 2.22 \cdot 10^6$ for a mobility of 1.5 cm$^2$ V$^{-1}$s$^{-1}$. According to these numbers, 1% of 1 nm particles and 5% of 5 nm particles are singly charged.

Although the data inversion assumes that the sampled particles are in charge equilibrium, the unipolar charger does not neutralize the aerosol sample entering the NAIS (McMurry et al., 2009). Thus, if the sample is highly overcharged, this can lead to overestimation of the ion and particle concentrations

### 3.6.4 CRITICAL: Assumption of singly charged ions

In the ion mode, the inversion considers only an aerosol particle mobility. Therefore, it produces a mobility distribution which user will later convert into a size distribution assuming that all the detected ions are singly charged. A direct mobility-size conversion will miss the possibility of multiple charges. To simplify, in ion mode we make an assumption that all charged particles are singly charged. In practice, this means that ion concentrations in the size range from ~20 nm to 40 nm may be overestimated and the shape of the distribution may be distorted.

In addition, Alguacil and Alonso (2006) reported that when using corona discharging a substantial fraction of doubly charged particles occur for particle diameters down to 15-20 nm. The measurement uncertainty of the NAIS increases above 20 nm because the corresponding electrometers are also affected by multiply charged particles with diameters up to 90 nm. Due to the limited mobility the range of the DMA, the data inversion cannot completely account for the effect of those particles. Therefore, we recommend that the ion and particle number size distributions above 20 nm should not be used without a correction. One possibility would be to merge the number size distribution measured with the NAIS into a distribution measured with e.g. Differential Mobility Particle Sizer (DMPS, Wiedensohler et al., 2012) in size range from 10 to 1000 nm to obtain information on the background aerosol population to be used in the data inversion (e.g. Kulmala et al., 2012).

### 3.6.5 Instrument function: transfer functions and loss estimation

Detailed description of a mathematical model of the NAIS in Mirme and Mirme (2013). The instrument response of the NAIS is a set of electric currents that are generated by the flux of ions precipitating on the collecting electrodes. An ion mobility distribution $f(z)$ is linked with the electrometer currents $y_i$ ($i = 1 \ldots$ n) using the analyser response function $G(i, z)$.

$$y_i = \int [eG(i,z)] f(z) dz, (i = 1, \ldots, n),  \qquad (7)$$



where $e$ is the elementary charge.

The analyser transfer function $G(i, z)$ is the response of electrometer $i$ to singly charged ions with mobility $z$. The function is derived in a straightforward way (Mirme and Mirme 2013) and it is verified by calibration measurements in ions mode. The diffusion losses for sampled particles inside the instrument are estimated by fitting the diffusion length parameter of the instrument model to ion mode calibration results. The diffusion and electrostatic losses in the sampling lines prior to the instrument should be taken into account by the user.

During particles measurements the corona charger is active. The instrument response in particles mode additionally includes the charging probability function $P(q, r)$.

$$y_i = \int \left[ e \sum_q q \, G(i, z(r, q)) P(q, r) \right] f(r) dr, (i = 1, \ldots, n), \qquad (8)$$

where $f(r)$ is the particle size distribution and $P(q, r)$ is the probability that a particle with radius $r$ carries $q$ elementary charges.

The analyser transfer function $G(i, z)$ is identical for both ions and particles measurements. The charging probability function $P(q, r)$ is based on a theoretical charging model. The function is adjusted and verified using calibration measurements in particles mode.

The data inversion finds an approximate ion mobility distribution $f(z)$ or particle size distribution $f(r)$ that best satisfies Eq. (7) or (8). The distributions are estimated as a sum of predefined elementary distributions $F_i(r)$.

$$f(r) = \sum_j \phi_i \cdot F_i(r), \qquad (9)$$

This allows to transform Eq. (7) or (8) to an equivalent matrix form.

$$y_i = \sum_j H_{ij} \cdot \phi_j, (i = 1, \ldots, n), \qquad (10)$$

where $H$ is the instrument matrix and $H_{ij}$ determines the response of the electrometer $i$ to the predefined distribution $F_i(r)$. The data inversion procedure solves the matrix Eq. (10) for the spectrum vector $\phi_j$ and calculates the size or mobility distribution estimate using Eq. (6).

The ratio of sample flow to total analyser flow is about 1:3 for NAIS which is quite large and even perfectly monomobile particles will have a response on several electrometers. For particles with diameters above 25 nm the probability of acquiring more than one elementary charge in the corona charger is non-negligible. Hence the electrometer response for larger particles becomes even wider. This also means that the measured electrometer signal may be a combination of by both singly charged smaller particles and multiply charged larger particles with the same mobility.

The multiply charged particles do not require special treatment in the data inversion. They are naturally included in the calculated response of the electrometers, i.e. the instrument matrix. However the uncertainty of the measurement results gradually increases for particle sizes above 25 nm because the electrometer responses become less distinguishable for larger particles and because a gradually larger portion of the response will be lost beyond the mobility range of the DMA.





**Additional particle losses.** The electrostatic losses inside the corona charger during charging process lead to underestimation of the particle concentration (Alonso et al., 2006; Huang and Alonso, 2011). The diffusion losses decreases and electrostatic loss increases as the charger voltage is increased, whereas charging efficiency increases with particle size and charger voltage. Electrostatic loss of small particles increases with decreasing particle diameter.

**3.7 Data processing**

To obtain the aerosol size distribution, $dN/d(log_{10}D_p)$, from the current signal caused by charged particles and the voltages applied in classification, the data inversion take into account the charge distribution of the aerosol particles, DMA flows rates, the DMA transfer function, the detection efficiency of the detector and losses in the instrument. In the ion mode, the Spectops inversion algorithm converts raw signal from 21 electrometers into 28 normalized mobility fractions, whereas in

the particle mode, the raw data is converted into 29 size fractions. In the data processing the user should do the following corrections and quality checks for the NAIS data.

**3.7.1 Data cleaning and quality check**

After the data collection and prior to the data analysis, the data should be quality controlled (i.e. processed). The bad data should be removed from the final data. Figure 16 illustrates some examples of typical faulty spectra and the removed parts of

the spectra. Data quality checks and criteria which should be fulfilled for the *ion number size distributions*: 1) negative and positive ion number size distribution agree visually (a similar distinct shape for number size distributions in both polarities), 2) size distribution has a continuous cluster ion mode visible in both polarities with a mode peak at ~1 nm, see Fig. 16b, 3) when looking at the negative ion size distribution the mean diameter of cluster ion mode should be slightly smaller compared to positive cluster ions (typically one channel difference between the polarities), 4) in number size distribution plots the

smallest of Aitken mode charged particles should be visible and have a diurnal variation at 25–42 nm size range, and 4) time series of total ion number concentration between polarities should agree within 10–20%, 5) same applies to small, intermediate ions and large ion concentrations, 6) when there is intermediate ions observed make sure that the size distribution doesn't have any gaps due to instrument malfunction, see Fig. 16d.

Data quality checks for the *particle number size distributions* are similar as for the ion data. In addition, check that 1) the

corona-charger generated ions do not dominate the particle spectra due to wrong post-filter settings. The 2–3 nm particle concentrations should stay in a range of ~200–700 cm-3 when no new particle formation event is not taking place, 2) determine the smallest detectable size using both positive and negative discharging (i.e. in both polarities) (Sect 3.7.4), and 3) select the preferred polarity to give the final data for measured particle number size distribution (Sect 3.7.5).

If possible the ion and particle data measured with the NAIS should be compared with supplementary data. Data quality

check for the offset mode: electrometer currents during offset measurements should not exceed ±10 fA.



### 3.7.2 Ion data: Converting from mobility to size distributions

In the ion mode, the Spectops inversion algorithm converts signal from 21 electrometer into 28 normalized ion mobility distributions, $dN/d(log_{10}Z_p)$, which the user has to convert into ion size distributions, $dN/d(log_{10}D_p)$. We recommend to do the mobility to diameter conversion using the Millikan-Fuchs equivalent mobility diameters introduced in Sect. 3.6.1.

To do this, the user should follow steps: 1) Open the spectra data files to get the geometric means of all 28 mobility fractions and calculate the lower and upper mobility limits for each mobility fraction. 2) Calculate the $d(log_{10}Z_p)$ for all mobility fractions using these limits. 3) Calculate next the absolute number concentration for each mobility fraction starting from the normalized ion mobility fractions: $dN/d(log_{10}Z_p)*d(log_{10}Z_p)=dN$. 4) Calculate the corresponding mobility diameters for each mobility fraction limits by determining the lower and upper limiting diameter. 4) Calculate the $d(log_{10}D_p)$ for all size

fractions. 5) Normalize the absolute concentrations using the determined size fractions: $dN*1/d(log_{10}D_p) = dN/d(log_{10}D_p)$.

### 3.7.3 CRITICAL: Correction for diffusional losses at sampling line

Diffusion losses inside the sampling lines should be taken into account by post▪processing of the data. Particle losses by diffusion in a straight line can be described by calculating a size-dependent particle penetration (Hinds, 1892). In laminar flow, these losses depend only on the line length, the flow rate through the line, and the particle size. In cases that bends

cannot be avoided in the sampling pipe, the size-depended particle penetration can be calculated according to Wang et al. (2002).

### 3.7.4 Particle data: Determining the smallest detectable size

In particle mode, raw data is typically converted into 29 particle size distributions, $dN/d(log_{10}D_p)$, by the Spectops software. During the particle mode measurements, the corona▪generated ions complicate the particle detection as the measurement size

range overlaps the size range of the charger ions. The positive and negative corona▪generated ions are smaller than 1.8 and 1.6 nm, respectively, which results in the lower detection limit of approximately 2 nm for the NAIS particle measurements (Manninen et al., 2012). Therefore, particles below a diameter of about 2 nm cannot be reliably distinguished from the corona▪generated ions. The lowest detection limit for the NAIS in the particle mode is between 2 and 3 nm depending on the corona voltage and on the properties and composition of carrier gas (environmental conditions). It is important to notice that

the lowest detection limit of the NAIS varies according to the current environmental conditions. The lowest detectable size of the NAIS in particle mode should be checked on regular basis, at least separately for different seasons.

### 3.7.5 Particle data: Selecting preferred polarity of corona discharging

Due to the design of the instrument, the particle spectra (i.e. particle number size distribution) is measured with both positive and negative corona charging. Ideally the distributions should be identical. A large and persistent difference may indicate a

problem with the measurements. We recommend to give only one particle spectra in the final processed data to avoid



misunderstandings. Thus, the user needs to decide on the preferred polarity on the particle data which is reported as the final particle number size distribution. Typically, the preferred polarity is chosen with two main criteria: 1) The polarity which has lower background level of the corona-charger ions extending to 2–3 nm size range (which is considered the lowest detection limit of the NAIS in particle mode). 2) The polarity which shows no short-time fluctuation over time (i.e. corona-charger ion

background stays same level over the diurnal cycle). Small corona-charger ion background ($\sim 10^2$) should be visible at 2–3 nm to be sure that the particles in aerosol sample are not filtered with the electric post-filter.

## 4 Troubleshooting

The troubleshooting, in Table 2, lists how to recognize faulty spectrum, potential problems and their solutions ranging from the instrumental to software issues. For further NAIS problem solving and identifying symptoms while operation see

Supplement S2.

## 5 Anticipated results

The list of locations and altitudes where frequent aerosol particle formation has been observed is still growing as new measurement campaigns are organized and field sites established (reviewed in Hirsikko et al., 2011).

### 5.1 Direct observation of atmospheric particle formation and growth

The ion spectrometer measurements performed within the EUCAARI project (Kerminen et al., 2010; Kulmala et al., 2011) present, so far, the most comprehensive effort to experimentally characterize nucleation and growth of atmospheric molecular clusters and nanoparticles at ground-based observation sites on a continental scale (Manninen et al., 2010). The atmospheric particle formation data analysis routines for the NAIS data, e.g. estimating the contribution of ions to particle formation, calculating the cluster ion and aerosol particle formation and growth rates, and the ion-ion recombination rates, is

described in details in separate procedure article (Kulmala et al., 2012).

### 5.1.1 Cluster ions, intermediate ions, and large ions i.e. charged particles

Charged particles are divided into small ions (1.3–0.5 cm$^2$ V$^{-1}$s$^{-1}$), intermediate ions (0.5–0.034 cm$^2$ V$^{-1}$s$^{-1}$), and large ions (0.034–0.0042 cm$^2$ V$^{-1}$s$^{-1}$), which correspond to mobility diameters of 0.8–2 nm, 2–7 nm and 7–20 nm, respectively (Hõrrak et al., 2001). As can be seen from Fig. 17, the ion number size distribution and time series of ion concentration measured

with the NAIS has a distinct shape during new particle formation. The concentration of small air ions in the atmosphere is determined by competition between production and loss processes (e.g. Israël, 1970; Hirsikko et al., 2011). Small ions i.e. cluster ions are detected in all environmental conditions where they have been measured with an ion spectrometer varying from extremely polluted areas (e.g. Backman et al., 2012; Herrmann et al., 2014) to extremely clean environments (Virkkula




et al., 2007) as well as from the lower troposphere to the free troposphere (e.g. Manninen et al., 2010; Mirme et al., 2010; Rose et al., 2015). The only exception where no cluster ions were observed is when measured inside cloud (Lihavainen et al., 2007). On the other hand, the intermediate ions are a strong indicator for secondary aerosol formation in various environments (Dos Santos et al., 2015; Leino et al., 2015), whereas the large ions represents the naturally charged fraction of

Aitken and accumulation mode particles.

Ionisation of air molecules (e.g. $N_2$ and $O_2$) produces primary air ions: positive ions and free electrons. The primary ions undergo rapid chemical reactions, getting neutralised and charged again, and become small ions in less than a second from their formation. To avoid misunderstandings, it should be noted that these primary ions are not detected by the NAIS as their electrical mobility is too large from classification and their lifetime is too short for the detection with the current version of

the instrument.

### 5.1.2 Neutral clusters and total aerosol particles

The lowest detection limit for the NAIS in the particle mode is approximately 2 nm due to overlapping corona-charger ions (Asmi et al., 2009; Manninen et al., 2011). Thus, the NAIS is not able to detect the pool of stable neutral clusters at sub-2 nm (Kulmala et al., 2013). The NAIS can detect only a "shoulder" of this neutral cluster pool. The NAIS in a very capable tool

for detecting the newly formed particle already at the 2-3 nm size range depending on the post-filter settings. The time series of 2-3 and 3-6 nm particle concentrations on new particle formation day are shown in Fig. 18. In particle mode, the NAIS overestimates the total particle number concentrations by a factor of 2–4 (Manninen et al., 2009; Gagné et al., 2011). The quantitative agreement improves at conditions representing particle formation bursts when higher particle concentrations are typically observed in the overlapping size range (nucleation mode). As seen in Fig. 19, merging particle number size

distributions measured with the NAIS (2.5–40 nm) and a differential mobility particle sizer (DMPS, 40–1000 nm) without any fitting highlights the issue at 20–40 nm range where the agreement is poor, as the NAIS overestimates particle concentrations. Therefore, we recommend to use the particle spectra measured with the NAIS up to 20 nm.

### 5.1.3 Contribution of ions to aerosol processes

Neutral particle formation seems to dominate over ion-induced and ion-recombined nucleation, at least in the continental

boundary layer (Lovejoy et al., 2004; Manninen et al., 2009, 2010; Zhang et al., 2012; Kulmala et al., 2013). The results obtained from the NAIS particle and ion measurements agree well with separate independent measurements performed with other electrical mobility spectrometer (e.g. Gagné et al., 2010, 2011) and condensation based techniques (Lehtipalo et al., 2009, 2010; Kulmala et al., 2013; Rose et al., 2015). Atmospheric ions participate in the initial steps of new particle formation, although their contribution was minor in the boundary layer (e.g. Kulmala et al., 2013). The highest atmospheric

particle formation rates are observed at the most polluted sites where the role of ions was the least pronounced (Manninen et al., 2010). Furthermore, the increase of particle growth rate with size suggests that enhancement of the growth by ions was negligible (Yli-Juuti et al., 2011).





## 5.2 Typical number size distributions in different environments

It can be noted from Fig. 20 that typical atmospheric ion and particle distributions measured with the NAIS varies much from a regional new particle event day to very clean day when the cluster ions are the most dominant feature in the ion distribution between 0.8 and 42 nm. A closer look at the particle formation, in Fig. 20f–h, reveals that the nucleation bursts

are usually observed during daytime and mostly starting before noon. Based on the visual shape of the time series of the number size distribution, several nucleation event types have been characterized (Hirsikko et al., 2007; Manninen et al., 2010).

## 5.3 Additional value of the NAIS measurements

### 5.3.1 Parametrized ion and particle formation and growth processes

By conducting measurements according to the procedure, it is possible to develop simple yet sufficiently accurate nucleation parameterizations for large-scale atmospheric modelling. Secondary aerosol formation includes the production of nanometer-sized clusters from atmospheric vapours and the growth of these clusters to larger particles. One dynamic process modifying the size distributions of neutral and charged clusters is ion–ion recombination which was parametrized by Kontkanen et al. (2013). Nieminen et al. (2009) derived a parameterization for ion-induced nucleation or, more precisely, for the formation

rate of charged 2-nm particles. In addition, it is important to predict nanoparticle growth accurately in order to reliably estimate the atmospheric cloud condensation nuclei concentrations. Häkkinen et al. (2013) introduced a semi-empirical parameterization for sub-20 nm particle growth that distributes secondary organics to the nanoparticles according to their size and is therefore able to reproduce particle growth observed in the atmosphere. All semi-empirical parametrizations described here are based on extensive NAIS data-sets to test how well the parameterization captures the seasonal cycle of the

modelled parameters and to determine the weighing factors in different environments. Leppä et al. (2009) introduced an aerosol dynamical box model which includes basic dynamical processes (e.g. condensation, coagulation and losses by deposition) as well as ion–aerosol attachment and ion–ion recombination. This model was validated and constrained by NAIS data.

### 5.3.2 Connection to atmospheric electricity parameters

Small ions are always present in the air and are responsible for the atmospheric electrical conductivity (e.g. Harrison and Carslaw, 2003). The early research of air ions was mainly focused on atmospheric electricity to study e.g. air quality (Israël, 1970). Tammet et al. (2009) suggested that air ions and the atmospheric electric field controlling the migration of ions should be considered when discussing the formation of primary and secondary particles. Air (polar) conductivity can be calculated directly from the ion number size distributions measured by the NAIS, in addition to reporting the concentration

of small, intermediate, and large ions, and the average small ion mobility.



## 6 Discussion and conclusions

We work towards better understanding the formation and growth mechanisms of aerosol particles using experimental observations and parameterized formation and growth mechanisms. The first steps to understand the role of ions and particles in global climate are to understand where, when and why nucleation mode particles are formed. The current level of

understanding the aerosol effects leads to large uncertainties in global climate model predictions (IPCC, 2013). Thus, the current aerosol process models need to be improved to capture the dynamics at sub-20 nm. To constrain and validate these models reliable field observations are needed. Here, we aim to provide tools to harmonize the measurements performed with the NAIS leading to comparable results which may be used for aerosol process parametrization and global model validation.

This work is part of the protocol work done within an ACTRIS community. The ACTRIS is a European Research

Infrastructure for the observation of Aerosol, Clouds, and Trace gases, and aims to serve e.g. a vast community working on models and forecast systems by offering high quality atmospheric data. Several large-scale modelling studies have demonstrated that more reliable nucleation parameterizations than currently available are needed to evaluate the importance of nucleation in climate (Spracklen et al., 2006; Makkonen et al., 2009; Merikanto et al., 2009; Pierce and Adams, 2009; Yu, 2010). Based on the NAIS results, nucleation parameterizations (e.g. size-dependent atmospheric nanoparticle growth and

nucleation favoring ion processes) already exist for large-scale modelling but no global model is using those (see Nieminen et al., 2011; Kontkanen et al., 2013; Häkkinen et al., 2013). Thus, the implementation of these parametrizations to models is the missing step to reach the full scientific understanding. Overall, the large goal is to integrate the NAIS to international research networks as a standard instrument to detect the atmospheric nanoparticles when studying climate and air quality, and to increase the visibility of extensive NAIS data sets (submission to international databases and determining standardised

data format).

We believe that by improving the accuracy and comparability of the measurements and instrument's laboratory characterization, we also improve the fundamental understanding of the physical processes measured by the ion spectrometers.



**Team list**: Dr. Hanna E. Manninen, Dr. Sander Mirme, Dr. Aadu Mirme, Prof. Tuukka Petäjä, and Prof. Markku Kulmala

**Data availability:** All data necessary for a reader to understand and evaluate the conclusions of the paper are included in the paper or its supplement or will be archived in an approved database and made available to any reader. The ion number size distribution data presented in the figures is stored at the EMEP database (http://ebas.nilu.no).

**Code availability:** All codes necessary for a reader to understand and evaluate the conclusions of the paper will be archived in an approved database and made available to any reader.

**Supplement link (will be included by Copernicus)**

**Author contribution:** H. E. Manninen was primarily responsible for the design and interpretation of the reported experiments. S. Mirme and A. Mirme provided technical support, and T. Petäjä and M. Kulmala administrative and supervisory support that made a direct substantial intellectual contribution to this research. H. E. Manninen prepared the manuscript with contributions from all co-authors.

**Acknowledgements**

This work is based on the long-term experience by the University of Helsinki (UHEL) performing field, laboratory and chamber measurements with the NAIS, and the SOP's written by the UHEL working group for the ACTRIS community (the European Union's FP7 capacities programme under grant no. 262254, and Horizon 2020 research and innovation programme under grant no. 654109). We acknowledge Dr. John Backman, Dr. Alessando Franchin, MSc. Janne Lampilahti, MSc. Katri Leino, and Dr. Ville Vakkari for their contribution on providing material for this SOP. UHEL acknowledge the Academy of Finland Centre of Excellence (grant no. 272041). H. E. Manninen acknowledges support by the Finnish Cultural Foundation (grant no. 00121082).

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





**Tables**

Table 1. Recommended measurement modes and measurement cycle of the NAIS.

|  | Ion measurements | | Particle measurements | |
|---|---|---|---|---|
|  | **Pos. ions** | **Neg. ions** | **Pos. charging** | **Neg. charging** |
| DMA polarity | positive | negative | positive | negative |
| Discharger | off | off | off | off |
| Filter | off | off | off | off |
| Charger | off | off | on, pos. HV | on, neg. HV |
| Post-filter | off | off | on | on |
| Duration (s) | 90 (45) | | 90 (45) | |

|  | Offset measurements | |
|---|---|---|
|  | **Pos. polarity** | **Neg. polarity** |
| DMA polarity | positive | negative |
| Discharger | on, neg. HV | on, pos. HV |
| Filter | on | on |
| Charger | off | off |
| Post-filter | off | off |
| Duration (s) | 30 (30) | |

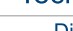
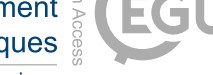

**Table 2. Troubleshooting.**

| Problem | Possible reason | Solution |
|---|---|---|
| **Particle/ion distribution issues:** | | |
| No ions observed.<br><br>Ion number size distribution spectra in Spectops screen is blue. | Instrument or inlet is not properly grounded or inlet is too long.<br><br>Offset measurements have a problem. | Make proper grounding using a metallic wire.<br><br>To test the inlet losses, remove the inlet temporarily (see Sect. 3.3.1). |
| The particle or ion number size spectrogram has a continuous or intermittent red stripe as a function of time at a certain size or mobility.<br>Or one of the electrometers is saturated continuously. | One or more electrometers are noisy or have high offset current due to deposited dirt or fibers inside the analyzer. | Clean the electrometer rings of that polarity where the problem is (Sect. 3.5.3).<br>*Note: Opening the mobility analyzer and cleaning it can end up making it more dirty if not done in a clean environment.* |
| No cluster (small) ions detected and a high concentration peak at around 5 nm. Otherwise ion spectra look normal. | The top part of the inner electrode of the DMA is short circuited to zero potential. No electric field to classify small ions. This happens when the isolation of one of the three centering rods of the "cartwheel" is damaged. | Remove the preconditioning unit. Open the top part of the DMA and check the isolation between the inner and outer electrode of the DMA. Isolation is created with a round metallic connector (check that 3 rubber rings around the wires are on their place).<br>*Note, during transportation and routine cleaning these rubber rings move easily.* |
| Number concentrations measured in positive and negative polarity are not corresponding (not matching, Sect. 3.2.2). | Airflows are not correct. Instrument might have a leak. | Check instrument software warnings related to sample and sheath flows. See the instructions for general "Airflow related issues" below. |
| | The preconditioning unit has a problem: cables disconnected or switched. | Check that all the plugs connect properly to the correct sockets in the preconditioning unit.<br>Check software warnings related to chargers and filters. |
| | The electric fields are not symmetric on both polarities. | Check DMA voltages (Sect. 3.1.2). |
| Whole particle spectra is continuosly red in Spectops screen with very high concentrations (~$10^5$-$10^6$).<br><br>The upper edge of the corona charger ions is larger than 3 nm in particle mode (Sect. 3.7.4). | Corona charger ions are not removed properly by the post-filter. | Adjust the post-filter voltage (Sect. 3.4.4). |



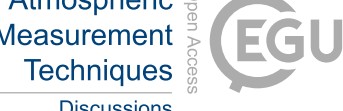

| | | |
|---|---|---|
| Ion and particle spectra in Spectops is red for few hours but instrument recovers itself. *Note: High concentrations more in negative polarity.* | Especially during rain and snow, some water might get inside instrument and deposit on electrometers. | Clean the instrument (see Sect. 3.5.1-3.5.2 and 3.5.5). |
| There are random vertical stripes in the spectrograms. | The central electrode voltage source is not stable. This induces simultaneous and identical fluctuations in many neighboring electrometers. | The central electrode voltage source needs repairs or replacement. |

| **Preconditioning unit issues:** | | |
|---|---|---|
| A discharger or charger current control voltage is near maximum. | A corona charger needle is dirty. The corona needle voltage is near maximum, but the charger current is still insufficient. | Clean or replace the corresponding corona needle (Sect. 3.5.5). |
| Charger current fluctuates or decreases rapidly. | Corona charger needle is dirty. | |
| A discharger or charger current control voltage is near minimum. | Preconditioning unit cables may be incorrectly connected. | Check that all the plugs connect properly to the correct sockets in the preconditioning unit. |
| | Leakage currents due to humidity prevent correct charger current measurement. | Run the instrument in dry conditions for at least 6 hours. |

| **Differential mobility analyser (DMA) issues:** | | |
|---|---|---|
| An analyser voltage is too low. | The voltages need 5 – 10 minutes to stabilise after power-on. | Wait for the instrument to warm up. |
| | An analyser voltage source is broken | Disconnect the corresponding analyser voltage plug and measure the voltage with a separate voltmeter. Contact Airel Ltd. |
| An analyser voltage is too high. | The voltage sources may drift slowly as they age and need readjustment. | Contact Airel Ltd. and send the power source for readjustment. |

| **Electrometer issues:** | | |
|---|---|---|
| An electrometer current is missing in the electrometer signal's table in Spectops screen. | The electrometer is constantly saturated due to a dirty electrometer. | Clean the analyzer (see Sect. 3.5.3). |
| The raw signal of an electrometer is zero in all operating modes while neighboring electrometers show signal. | The electrometer has a bad contact to the electrode. | Take the electrometer out and put it back. *Note: Some instruments have the test peaks screwed to the analyser and this issue is not relevant.* |



| | The electrometer has a bad connection to the data acquisition system. | Take the electrometer out, disconnect and reconnect the plug, put the electrometer back. |
|---|---|---|
| | The electrometer is broken. | Replace the electrometer.<br>To confirm the issue, swap the electrometer with another in the same instrument. |

| **Airflow issues:** | | |
|---|---|---|
| Instrument inlet and outlet flow are not equal. | Instrument has a leak.<br><br>*Note: Large leaks you can feel with a wet finger or use liquid leak detectors (e.g. Snoop Liquid Leak Detector, Swagelok) if leak is on overpressure side. Single, small leaks are impossible to detect.* | Check all the tubing and connectors for a leak. If needed open the whole instrument and reconnect all parts carefully. Pay attention to the O-ring seals.<br>Also see the instructions for general "Airflow related issues". |
| The blower is making a loud noise in the 1-blower system or the blower speed is unstable or decreasing slowly (>100 ccm/s in a day). | Blower stops soon working. | Replace the blower (in the 1-blower systems). See Sect. 3.5.4. |
| A flow control voltage is near maximum. | A blower is unable to provide the required flow rate or a flow sensor is incorrectly connected or broken. | See instructions for general "Airflow related issues". If everything else is ok the blower may be near end of life and require replacement. |

| **Software issues:** | | |
|---|---|---|
| The particle and ion spectra is not updated to the Spectops screen. | Spectops displays only the selected spectra. | Click "Show latest" icon on the task bar. |
| After installing the sofware to a new computer, an unidentified communication error forbids the measurements. | Spectops is not able to connect to the NAIS. | COM-ports greater than 9 need \\.\ prepended e.g. write "\\.\COM12". |



## Figures

a)

b)

c)

5   Fig. 1. Schematics of the NAIS with: a) 1-blower, b) 3-blower and b) 4-blower system.





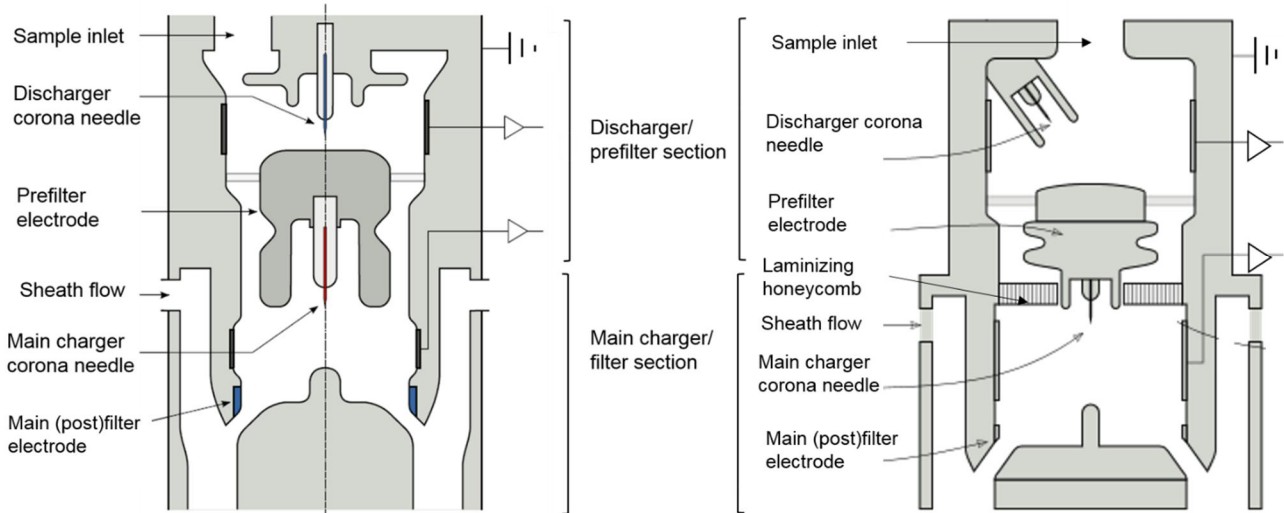

Fig. 2. A cross section of the aerosol preconditioning unit and upper part of the mobility analyser for old generation (left column) and new generation (right column) of the NAIS. Grounding of different parts visualized.





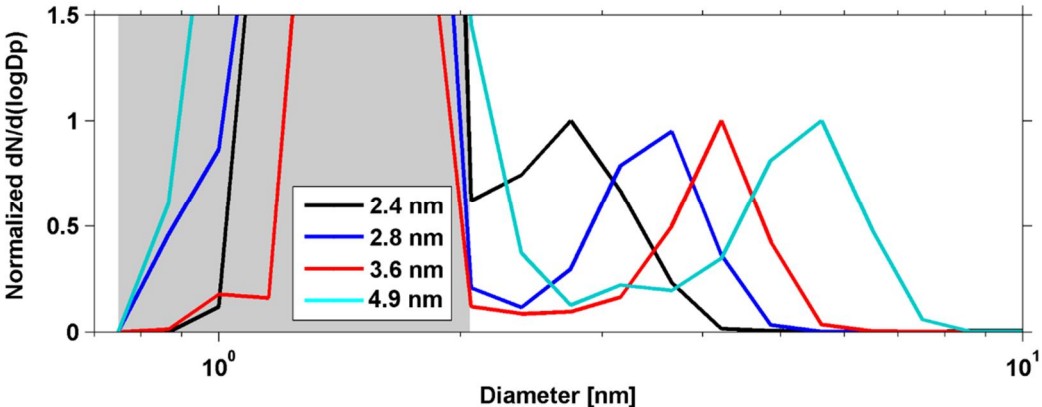

Fig. 3. The size distributions of four different sizes of neutral silver particles measured with the NAIS (positive charging, automatic post-filetring). The shaded area represents the size range of corona charger ions.





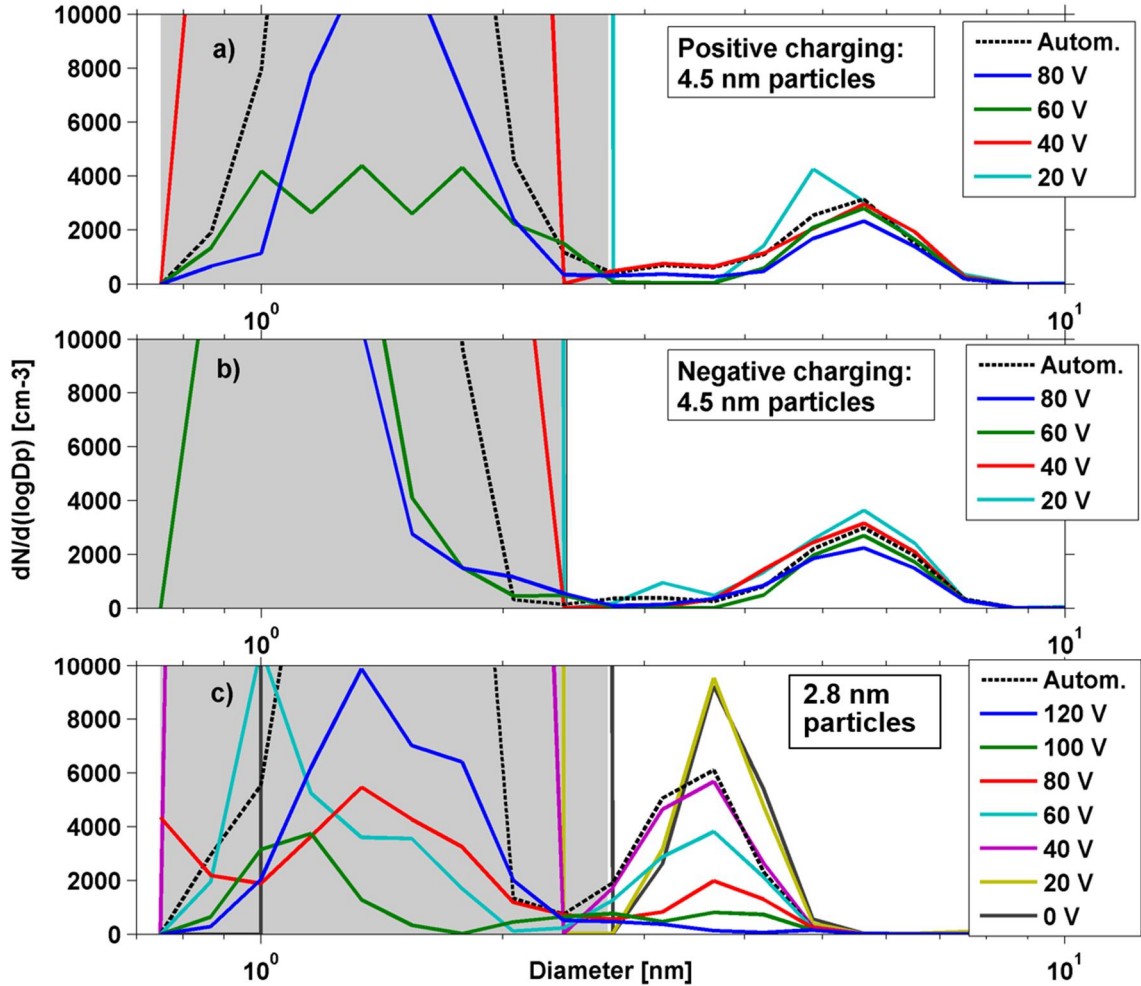

Fig. 4. The size distribution of 4.5 nm neutral silver particles measured with the NAIS using a) positive and b) negative corona charging, and c) the size distribution of 2.8 nm particles measured with positive charging. Different lines represent used post-filtering voltages.



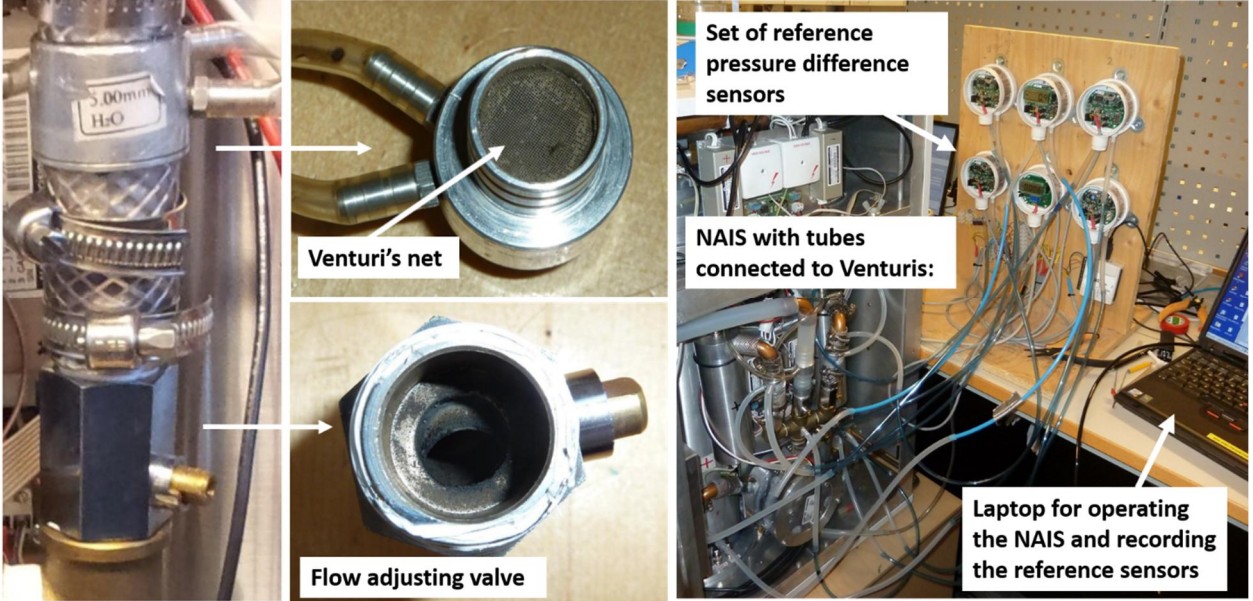

Fig. 5. On left, a dirty Venturi tube and its net, and a dirty flow adjusting valve in a close-up from Marikana, South Africa. Both need to be cleaned well before a flow verification. On right, an example of a set-up for flow checks with external pressure difference sensors which are connected to the 5 Venturi's to record all values simultaneously, including the pressure difference over the blower.





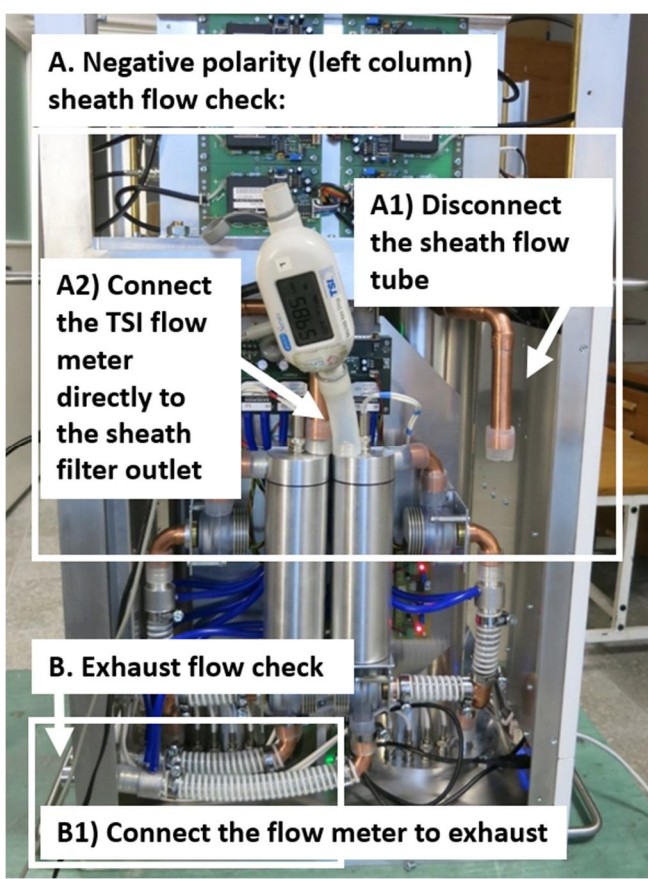

Fig. 6. Quick and easy sheath flow check for 4-blower system using a TSI flowmeter which is placed directly after the sheath air filter.



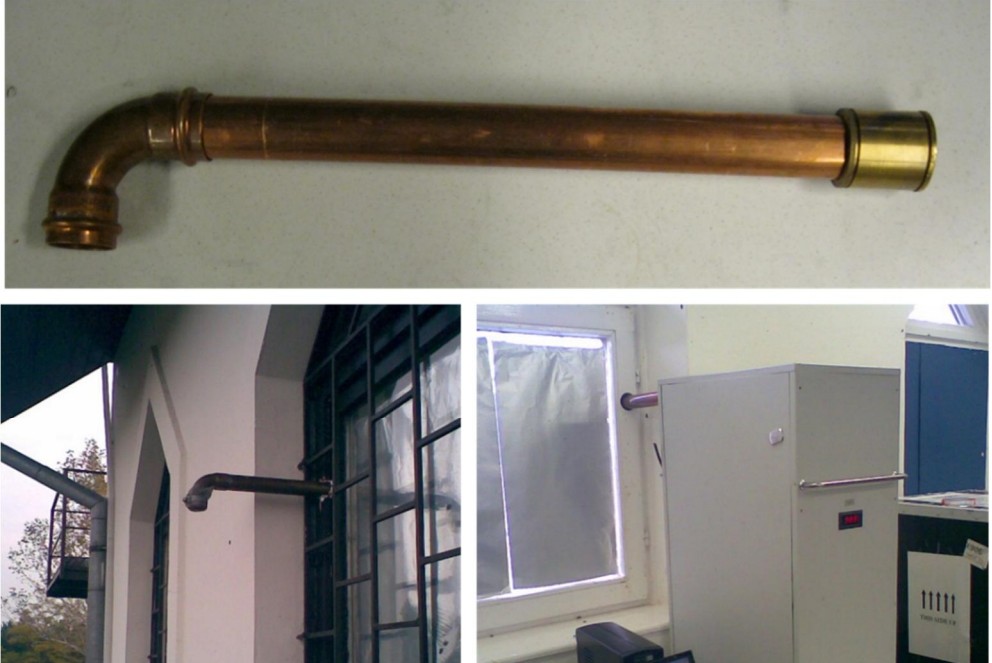

Fig. 7. Example an aerosol sampling inlet for the NAIS used at K-puszta, Hungary. A grid is missing in the top figure from the end of the sampling line. Below it is seen in an instrument set-up on field.



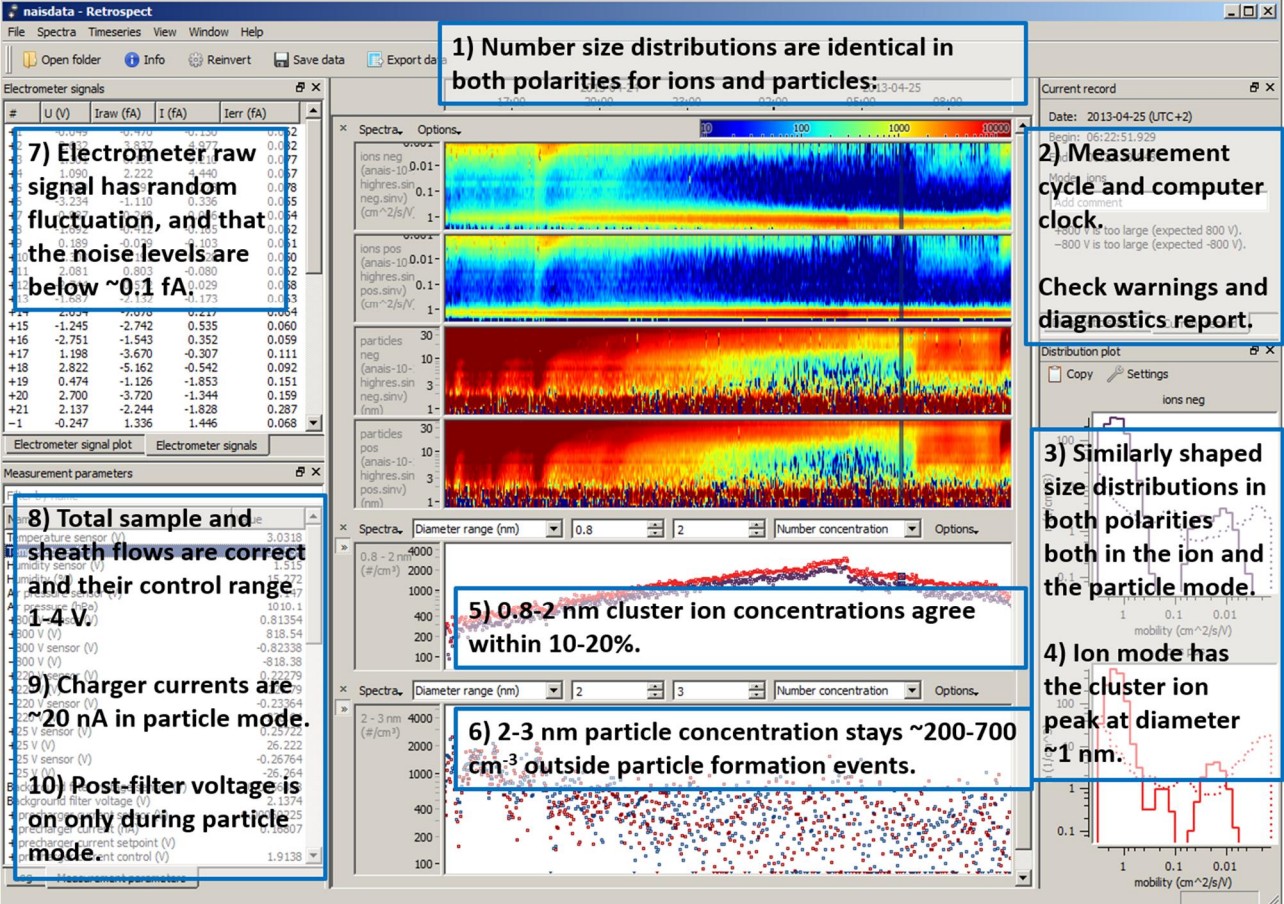

Fig. 8. The NAIS measurement software (Spectops.exe or Retrospect.exe) is recommended to be used for checking that the instrument is operating properly by following the steps listed in the figure.



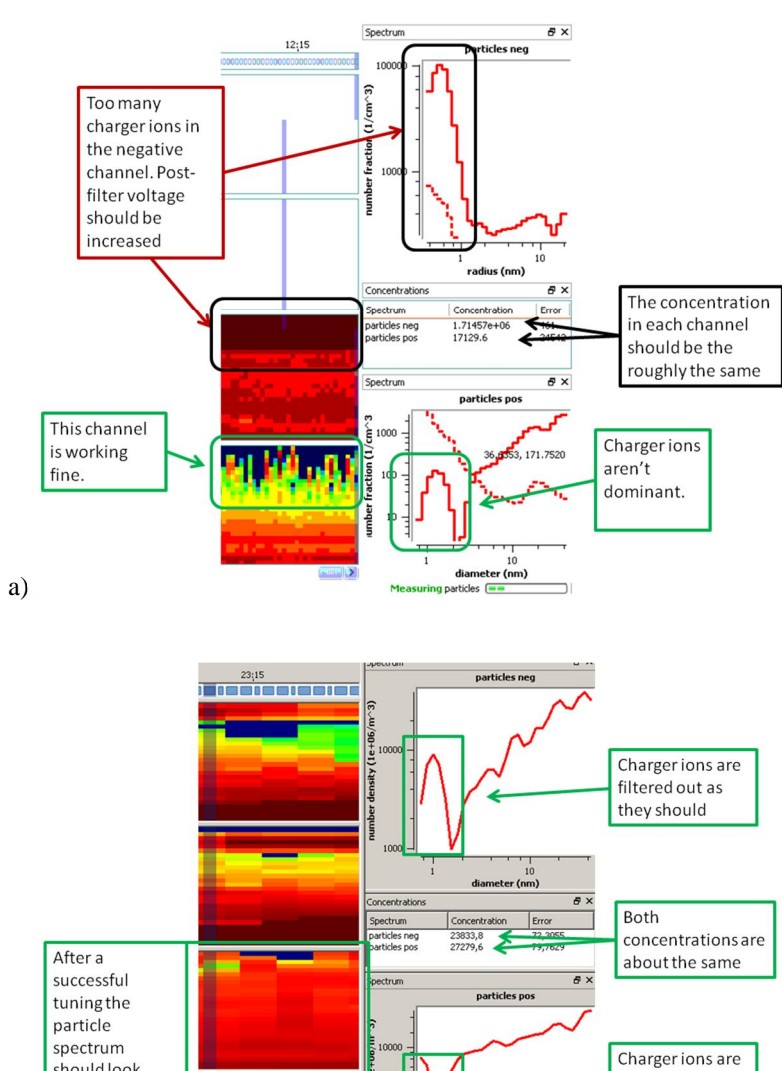

Fig. 9. Adjusting post-filter during the particle measurement mode: a) what to look for when tuning the post-filter voltage for a bad column (here negative polarity), and b) how the particle number size distribution should look like after tuning.



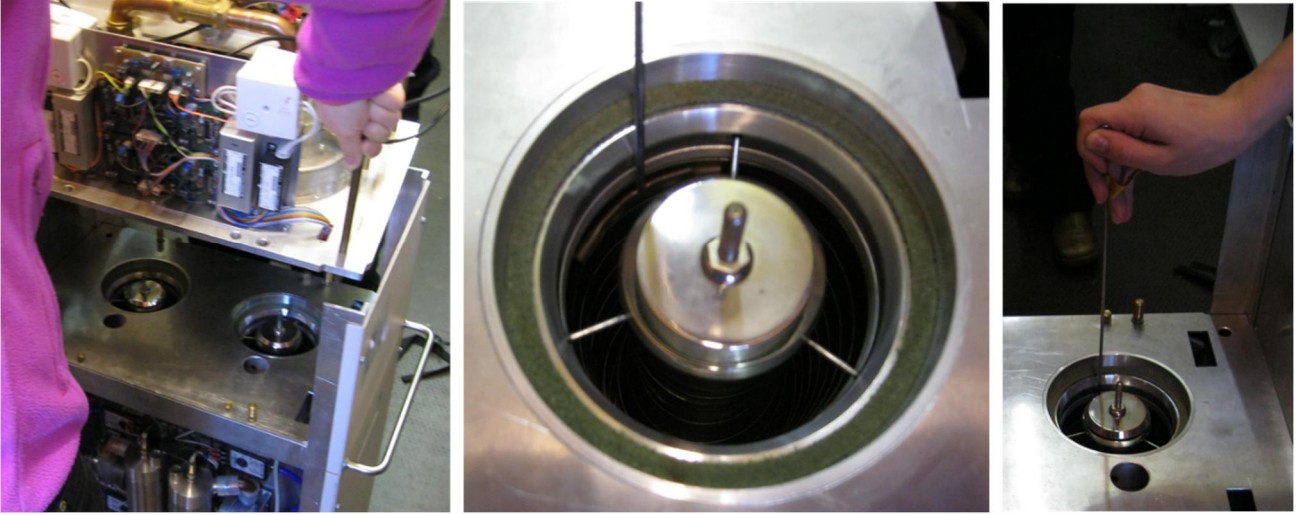

Fig. 10. Cleaning electrometer rings by lifting away the sample preconditioning unit (the top part of the NAIS) and using a long cleaning rod to wipe the surface of outer electrode.



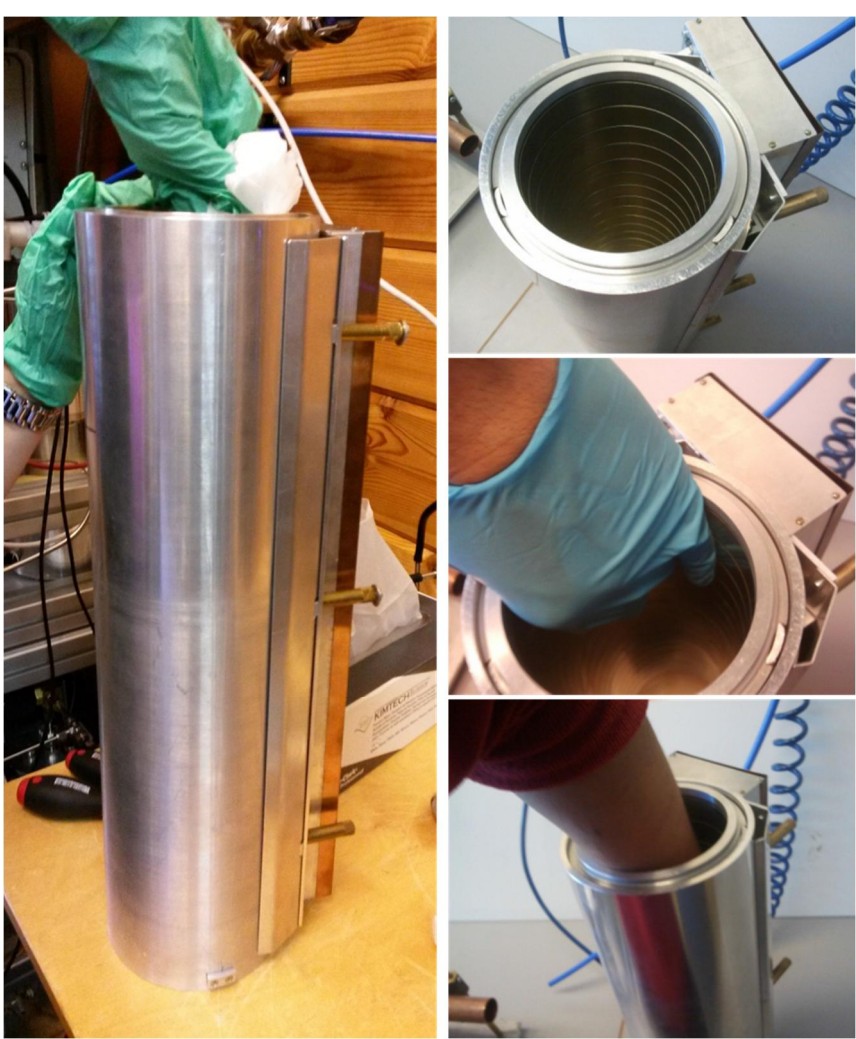

Fig. 11. Cleaning electrometer rings by opening the mobility analyser and wiping with clean cloth.





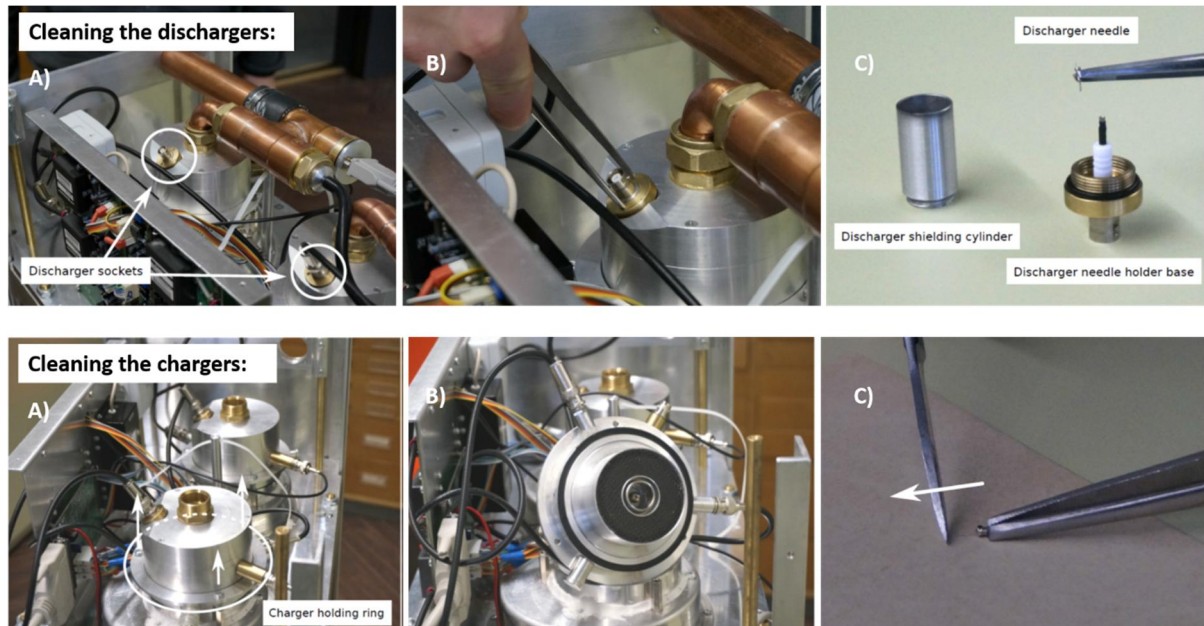

Fig. 12. Opening and cleaning the dischargers (upper row) and the chargers (bottom row).





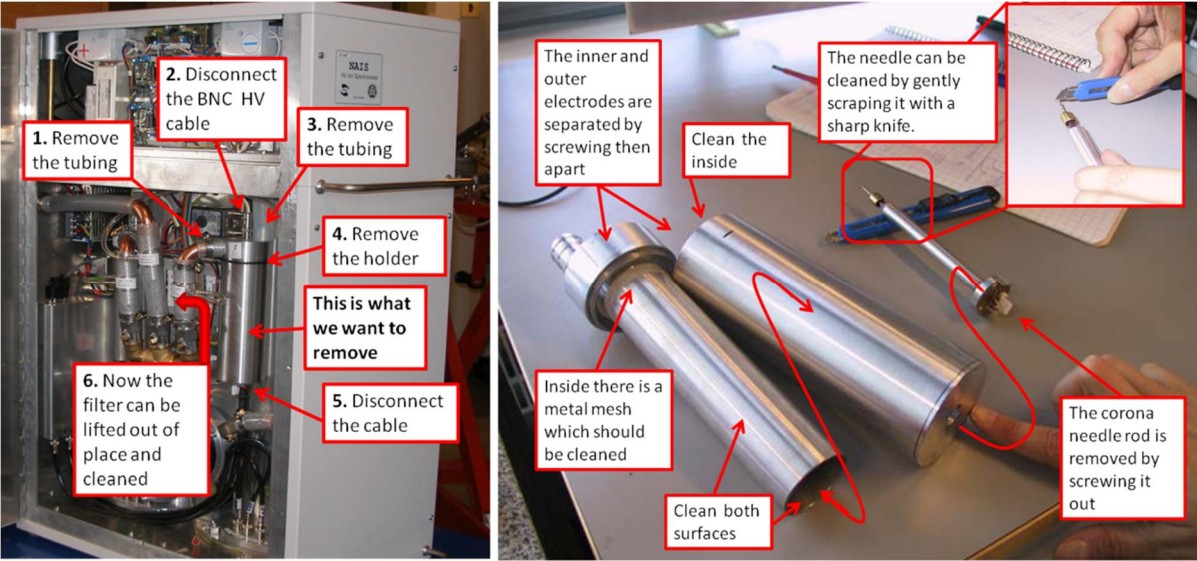

Fig. 13. Removing and cleaning the sheath air filters. Some of the newest NAIS's do not have corona needle inside the sheath air filter.



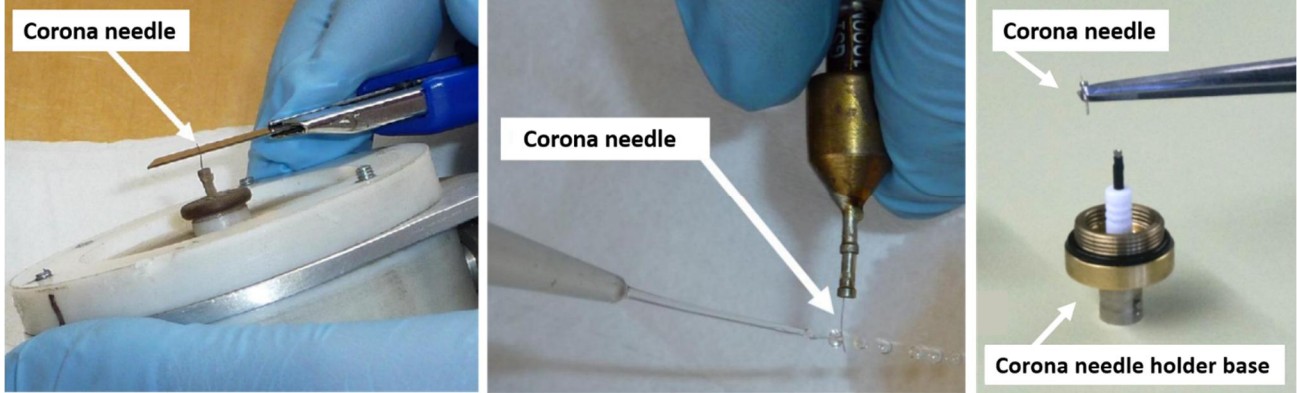

Fig. 14. Cleaning the corona needle with a sharp knife (left) or by rinsing with a dissolvent (middle), and replacing a corona needle using tweezers (right).



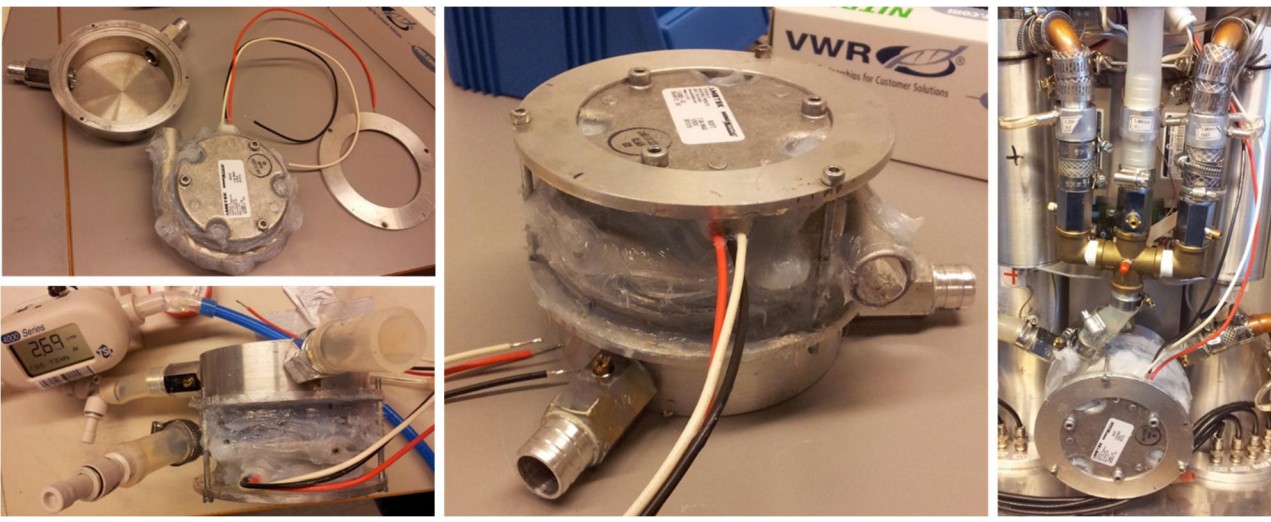

Fig. 15. Procedure for replacing a blower for a 1-blower instrument.





Fig. 16. Typical faulty ion number size distributions before (left column, problematic area boxed) and after (right column) cleaning and quality checks. The bad or missing data was selected and replaced with -999.99 using a MATLAB script. The faulty data was caused by a) continuously noisy electrometer, b) bad grounding at the top of the DMA (partly missing cluster ions), c) dirt in the bottom of the DMA (high noise), and d) classifying electric field inside the DMA not correct due to a bad BNC cable connection (intermediate ions missing).





Fig. 17. Example negative ion number size distribution during a new particle formation measured with the NAIS (upper panel), and concentration of negative cluster ions (0.8–2.0 nm nm), intermediate ions (2.0–7.0 nm), large ions (7.0–20 nm) and gas phase sulphuric acid (lower panel) on 5 May 2007 in Hyytiälä.



Fig. 18. Example particle size distribution measured with the NAIS in the size range 2.5–20 nm and with the DMPS in the size range 20–1000 nm on 5 May 2007 in Hyytiälä (upper panel), and concentrations of 2–3 nm particles measured with the

5    NAIS, 3-6 nm particles with the DMPS, and sulphuric acid (lower panel).





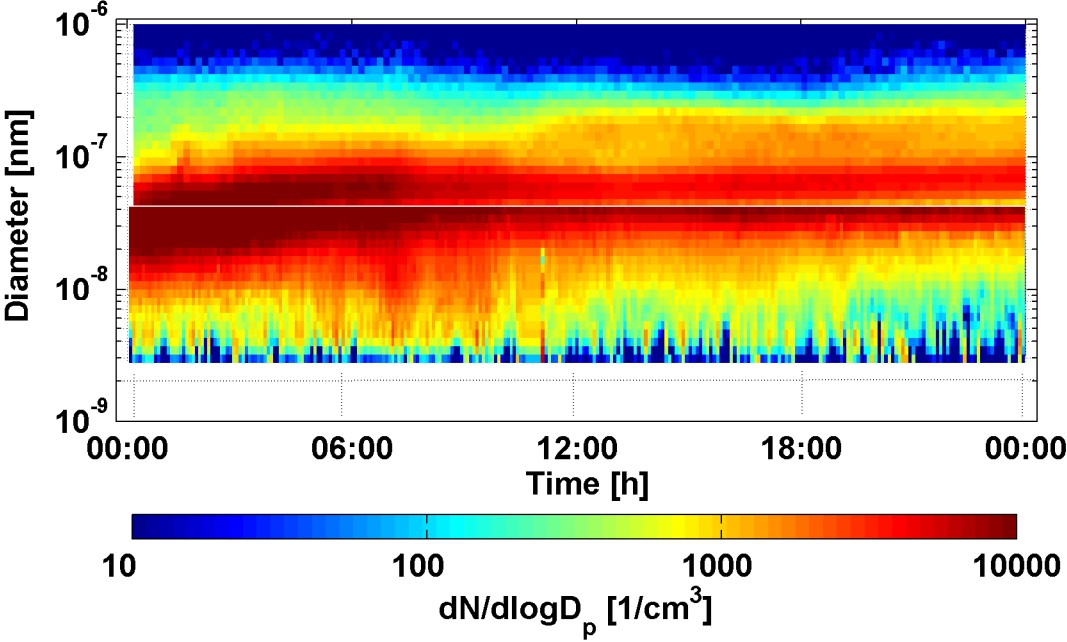

Fig. 19. Merging particle size distributions measured with the NAIS in the size range 2.5–40 nm and with the DMPS in the size range 40–1000 nm on 23 April 2007 in Hyytiälä.





Fig. 20. Typical negative ion number size distribution measured with the ion spectrometer in different environments and conditions a) no nucleation mode particles, just cluster ions observed in Finokalia on 17 Oct 2008, b) lower edge of Aitken mode particles and cluster ions obseved in Finokalia on 25 Oct 2008, c) intermediate ion bursts observed during heavy rain in Finokalia on 28 Dec 2008, d) intermediate ions observed during snow storm in Jungfraujoch on 28 Nov 2008, e) undefined particle formation in Cabauw on 15 Sep 2008, f) changes in air mass and particle formation in Melpitz on 3 Aug 2008, g) regional particle formation in Melpitz on 13 Apr 2009, and h) local new particle formation plume in Mace Head on 16 Oct 2008.