# Peer review of "How to reliably detect molecular clusters and nucleation mode particles with Neutral cluster and Air Ion Spectrometer (NAIS)"

_Atmospheric Measurement Techniques, 2016_

## Referee Comment (RC1) · Anonymous Referee #1 · 20 Apr 2016

The NAIS is an important instrument as it provides an insight into the physical processes occurring at the very sizes where atmospheric particles are formed. Unfortunately, since its development, there has been very little information on its operation and trouble-shooting procedures available in the literature. In that respect, I feel that this paper will prove to be invaluable. I have been using a NAIS for the last 4 years and I find that the answers to many of the questions that have arisen during this time are contained in this paper. The description of its operation and the many troubleshooting hints will be extremely useful to its many users. The authors also provide many useful tips on the interpretation of the data which I find to be extremely useful.

Furthermore, it is written by a group of authors who are best qualified to do so.

[Figure]

Therefore, I consider it to be a very important paper which fills a void that has been present for quite some time.

There are several typo's and grammatical errors that should be corrected. There are other possible minor improvements that could be made. Some obvious ones are listed below:

P 1, line 23: replace "happen" with "occur". P 2, lines 21-24 should be improved or replaced. I would suggest text similar to that used by Manninen et al (2009). P 3, line 11: Replace "descripted" by "described". And, perhaps delete "in a Nature protocols article". P 5, line 19: "It is recommended that ion spectrometers take part in the calibration and intercomparison workshops...". Not clear. Do you mean that "ion spectrometer users take part" or that "ion spectrometers should be brought and calibrated ... at ... " ? P 9, lines 15-16: "The instrument is operational both in vertical and horizontal position" – Do you mean the instrument or the inlet line? Can the instrument be placed horizontally? It continues as "However, the vertical orientation for inlet line is not recommended". Does this mean that the inlet line should not be placed vertically? Could you provide your reasons for this? Is it the electrode effect? "Recommended inlet height is 2 metres above the ground level". To my knowledge, several NAIS instruments are being used with the sampling tube out of upper floor windows and on the roofs of buildings. Is this not recommended? P 14, line 30: Replace "notice to" by "ensure that you". P 15, line 7: Replace "place" with "position". P 20, line 22 is very unclear. I would recommend, replacing "stay" with "remain" and delete either "no" or "not" from later in this sentence.

Table 2 P 34 Top Row: "inlet is too long". I think you mean that the "inlet tube is too long" ? Bottom Row: "Whole particle spectra is continuously red.." Do you mean "total particle" or "Both particle" ? Suggest "appears continuously red in colour" P 36 Airflow Issues: "Blower stops soon working". Do you mean "Blower soon stops working" ? But, I am not sure how this could be the "Possible Reason" for the issue. Is it not a consequence of the issue?
A comment on P 22, line 26: "The only exception where no cluster ions were observed is when measured inside cloud (Lihavainen et al., 2007)". While this is so, another clear example is shown in Jayaratne et al: Suppression of cluster ions during rapidly increasing particle number concentration events in the environment. Aerosol and Air Quality Research. 15, 28-37, 2015. They used a NAIS to monitor ions and particles during a local fireworks display and observed that, owing to the very high concentration of particles, the cluster ion concentration was suppressed to zero for a short time.

Some references are missing from the reference list. E.g. Alguacil and Alonso (2006), Alonso et al (2006), Huang and Alonso (2011),

Both Supplementary Sections are written in great detail and will, I am sure, be of great help to all present and future users of the NAIS. Just a minor suggestion – the Supplement giving the cleaning instructions could benefit with some improvement of the grammar.

The manuscript falls into the scope of AMT and the content is scientifically sound.

Following the minor revisions above, I recommend it for publication in AMT.

---

## Referee Comment (RC2) · Anonymous Referee #2 · 28 Apr 2016

**How to reliably detect molecular clusters and nucleation mode particles with Neutral cluster and Air Ion Spectrometer (NAIS)**

The paper by Manninen et al. aims at providing a standard operation procedure to NAIS users regarding both measurement and data processing. NAIS has already been widely used in a large variety of locations, giving significant insights into our understanding of the nucleation and growth processes. I clearly believe that persevering with this instrument using harmonized procedures would benefit to the whole community. This would especially allow more accurate analysis and comparisons of the datasets that can further serve for model investigations. For that reasons, I recommend the publication of this paper. However I have some comments that should be addressed in a revised version. Also, I suggest the paper to be read and cleaned up by a native English speaker.

**Specific comments**

P1, L20-23: "have remained unclear until very recently": can we really consider that the mechanisms and precursors involved in the particle formation process are now clear? I would balance this statement and rather say that our understanding of the formation process has benefited from direct atmospheric measurements that could be achieved by the improvement of measurement techniques.

P2, L13-14: The journal (Nature protocols) does not need to be specified in the text.

P4, L12: It should be mentioned that the recommended measurement cycle is for ground based measurement, since a different cycle is later suggested for airborne measurements (P11, L19).

P5, L1-5: If I understand well, the right bound of the shaded area on Fig.3 and Fig. 4 represents the lower detection limit of the NAIS? It is defined as the lower diameter for which post filtered concentrations can be considered negligible? This should be clearly stated since it is crucial for the user to be able to define this lower detection limit (see last comment regarding P21, L25-26).

P5, Section 3: The term "calibration" is used for different purposes with different meanings (e.g. P7, L2 :"the calibration (actually a flow verification)"). This can be misleading.

P6, L18-24: Can you give more details regarding the way to adjust the Venturis' pressure drops? This will also help for the 3- and 4- blower systems described on P7 (L21: "The valves should be adjusted accordingly.")

P7, L14: Can the calibration coefficient be directly adjusted by the user? If yes, can you give additional pieces of explanation? If no, clearly state that assistance from Airel is needed.

P10, L10-17: Can you precise the setup which is used for the "drying by heating". Should particular precautions be taken at altitude sites to limit RH in the sample flow? High RH conditions are especially found during in-cloud measurement and can coincide with very low temperatures.

P10, L26-27: Can you give more details regarding the way to take into account the deviation between target and actual flow rates in the processing of the number size distributions?

P11, L1-8: In case of measurements performed with a one blower system, are there corrections that can be made on the data afterwards to compensate for an unappropriated use of the blower?

P12, L21-22: You should precise that the full procedure is described in the supplementary.

P18, L22-26: I am not sure to understand what the authors mean when talking about "correction". Should we try to correct NAIS data based on SMPS measurements, and if yes how? Or should we just ignore NAIS data above 20 nm, as suggested on P23, L22?

P20, L29: "supplementary data". Be more precised.

P21, L12: Should corrections be applied even when using the recommended inlet manufactured by Airel?

P21, L25-26: In practice, how can we accurately find out the lowest detectable size given the fact that both residual generated ions and sampled particles are present at the smallest size ranges? This should be clearly explained using Fig. 3 and 4.

---

## Author Comment (AC1) · 2 Jun 2016

We would like to thank the referee for the encouraging review and the constructive comments and to help us to improve the manuscript. Below are our answers to the comments by the referee.
* * *
Answers to the referee comments by Anonymous Referee #1 on our manuscript "How to reliably detect molecular clusters and nucleation mode particles with Neutral cluster and Air Ion Spectrometer (NAIS)" by Hanna E. Manninen et al.

There are several typo's and grammatical errors that should be corrected. There are other possible minor improvements that could be made. Some obvious ones are listed below:

We revised the manuscript and corrected the language.

P 1, line 23: replace "happen" with "occur".

Done.

P 2, lines 21-24 should be improved or replaced. I would suggest text similar to that used by Manninen et al (2009).

Unclear comment. We don't understand which corrections the referee is suggesting. If the comment is related to a chapter few lines after starting with words "Special considerations", we have moved it to earlier location in the Introduction to fit the text better.

P 3, line 11: Replace "descripted" by "described". And, perhaps delete "in a Nature protocols article".

Done.

P 5, line 19: "It is recommended that ion spectrometers take part in the calibration and intercomparison workshops…". Not clear. Do you mean that "ion spectrometer users take part" or that "ion spectrometers should be brought and calibrated … at … " ?

Yes, the users should take part in the workshops. The ion spectrometers should be calibrated often enough, preferably at the calibration workshops.

We modified the text to make it clearer: *It is recommended that ion spectrometer users take part in the calibration and intercomparison workshops organized in co-operation by University of Helsinki and Airel Ltd. The ion spectrometers should be calibrated often enough, preferably at the calibration workshops. The goal is to organize these workshops on a regular basis. During the workshops the ion spectrometer flows are calibrated and their mobility classification and concentration measurements are verified."*

P 9, lines 15-16: "The instrument is operational both in vertical and horizontal position" – Do you mean the instrument or the inlet line? Can the instrument be placed horizontally? It continues as "However, the vertical orientation for inlet line is not recommended". Does

this mean that the inlet line should not be placed vertically? Could you provide your reasons for this? Is it the electrode effect?

We modified the text to answer referee's questions: *"Although the instrument and the inlet line can be placed vertically or horizontally, the horizontal orientation for the inlet line is recommended. In the vertical inlet setup the precipitation may easily enter the instrument and damage the instrument and lead to poor data quality."*

"Recommended inlet height is 2 metres above the ground level". To my knowledge, several NAIS instruments are being used with the sampling tube out of upper floor windows and on the roofs of buildings. Is this not recommended?

Very good point. Inlet sampling height depends on the surroundings. It can vary from 2 m above ground level to 15 m above ground level. The user does need to select the sampling site according to environment conditions.
    We modified the text accordingly: *"Sampling height depends on the surroundings. It can vary from 2 meters above ground level to 15 meters above ground level (height of the surrounding canopy/buildings)."*

P 14, line 30: Replace "notice to" by "ensure that you".

Done.

P 15, line 7: Replace "place" with "position".

Done.

P 20, line 22 is very unclear. I would recommend, replacing "stay" with "remain" and delete either "no" or "not" from later in this sentence.

Done.

Table 2 P 34 Top Row: "inlet is too long". I think you mean that the "inlet tube is too long" ? Bottom Row: "Whole particle spectra is continuously red.." Do you mean "total particle" or "Both particle" ? Suggest "appears continuously red in colour"

Both corrected as suggested above: "*… and inlet tube is too long*" and "*Total particle spectra appears continuously red in colour at Spectops screen with very high concentrations (~$10^5$-$10^6$).*"

P 36 Airflow Issues: "Blower stops soon working". Do you mean "Blower soon stops working" ? But, I am not sure how this could be the "Possible Reason" for the issue. Is it not a consequence of the issue?

We modified the text accordingly: "*The blower soon stops working as it is worn-out*".

A comment on P 22, line 26: "The only exception where no cluster ions were observed is when measured inside cloud (Lihavainen et al., 2007)". While this is so, another clear

example is shown in Jayaratne et al: Suppression of cluster ions during rapidly increasing particle number concentration events in the environment. Aerosol and Air Quality Research. 15, 28-37, 2015. They used a NAIS to monitor ions and particles during a local fireworks display and observed that, owing to the very high concentration of particles, the cluster ion concentration was suppressed to zero for a short time.

We added citation to paper: Jayaratne, R. E., Ling, X., and Morawska, L.: Suppression of cluster ions during rapidly increasing particle number concentration events in the environment. Aerosol and Air Quality Research, 15(1), 28-37, 2015.
     And modified the manuscript accordingly: *"The only exception where no cluster ions were observed is when measured inside a cloud (Lihavainen et al., 2007) or during a rapid, extreme increase in particle number concentration (Jayaratne et al. 2015)."*

Some references are missing from the reference list. E.g. Alguacil and Alonso (2006), Alonso et al (2006), Huang and Alonso (2011).

We added following papers to reference list:

*Alguacil, F.J. and Alonso, M.: Multiple charging of ultrafine particles in a corona charger, J. Aerosol Sci., 37, 875–884, 2006.*

*Alonso, M., Martin, M.I., and Alguacil, F.J.: The measurement of charging efficiencies and losses of aerosol nanoparticles in a corona charger. J. Electrostat., 64, 203-214, doi: 10.1016/j.elstat.2005.05.008, 2006.*

*Huang, C-H, and Alonso, M.: Nanoparticle electrostatic loss within corona needle charger during particle-charging process. J. Nanopart. Res., 13: 175-184, 2011.*

Both Supplementary Sections are written in great detail and will, I am sure, be of great help to all present and future users of the NAIS. Just a minor suggestion – the Supplement giving the cleaning instructions could benefit with some improvement of the grammar.

We will revise the supplements and correct the language.

---

## Author Comment (AC2) · 2 Jun 2016

We would like to thank the referee for the review and the constructive comments and to help us to improve the manuscript. Below are our answers to the comments by the referee. The page and line numbers used here refer to the original AMTD manuscript published online.

Answers to the referee comments by Anonymous Referee #2 on our manuscript "How to reliably detect molecular clusters and nucleation mode particles with Neutral cluster and Air Ion Spectrometer (NAIS)" by Hanna E. Manninen et al.

REFEREE COMMENT: For that reasons, I recommend the publication of this paper. However I have some comments that should be addressed in a revised version. Also, I suggest the paper to be read and cleaned up by a native English speaker.

**Our reply: We revised the manuscript and corrected the language.**

**Specific comments**

P1, L20-23: "have remained unclear until very recently": can we really consider that the mechanisms and precursors involved in the particle formation process are now clear? I would balance this statement and rather say that our understanding of the formation process has benefited from direct atmospheric measurements that could be achieved by the improvement of measurement techniques.

True. We modified the sentence accordingly: "Insight into the detailed formation mechanisms and the chemical composition of vapors, which participate in the atmospheric particle formation processes, has clearly benefited from direct atmospheric measurements and improvements in measurement techniques (Manninen et al., 2010; Kulmala et al., 2013, 2014; Ehn et al., 2014)."

P2, L13-14: The journal (Nature protocols) does not need to be specified in the text.

**We removed "... in a Nature protocols article" from the manuscript.**

P4, L12: It should be mentioned that the recommended measurement cycle is for ground based measurement, since a different cycle is later suggested for airborne measurements (P11, L19).

**Done. The manuscript was modified: "*Recommended measurement cycle* for the NAIS ground-based measurements is alternating between offset, ions and particle modes as follows: [offset 30, ions 90, particles 90], where the numbers represent measurement times in different modes in seconds."**

P5, L1-5: If I understand well, the right bound of the shaded area on Fig.3 and Fig. 4 represents the lower detection limit of the NAIS? It is defined as the lower diameter for which post filtered concentrations can be considered negligible? This should be clearly stated since it is crucial for the user to be able to define this lower detection limit (see last comment regarding P21, L25-26).

Yes, the gray shaded area represents the size range of corona charger-generated ions. In Fig. 3, the charger ions were clearly smaller than 2 nm. Fig. 4 shows that the corona-generated ions are slightly smaller when negative charging is used. The corona ions should be always

cut out of the particle spectra during the subsequent data processing. The lowest detection of the NAIS is determined as the upper edge of the gray shaded area which is equal to the upper edge of the corona charger ion size distribution. The determination of the lower detection limit should be done always when new particle formation (i.e. natural cluster activation and growth) is not taking place. To make sure that the lowest detection limit is at preferred size range, the post-filters should be adjusted so that the 2–3 nm particle concentrations should remain in a range of ~200–700 cm-3 including the corona generated ions in that size range.

We modified the manuscript by dividing this reply into two parts.

To comment P5, L1-5 we clarified the information on the two figures: "The corona charger ions have a mobility diameter range of  $1.0-1.6 \text{ nm} (1.3-0.8 \text{ cm}^2 \text{V}^{-1} \text{s}^{-1})$ , Manninen et al., 2011). These sizes define the absolute lower detection limit of the NAIS in the particle mode. The size range of corona charger-generated ions measured by the NAIS is illustrated in Fig. 3 with a gray shaded area. The figure shows that the charger ions were clearly smaller than 2 nm. As can be seen from Fig. 4, the electrical filtering of the charger ions and the inability to remove all the naturally charged particles plays an important role in determining the lowest detection limit to approximately at 2 nm in electrical mobility equivalent size. A post-filter is used to cut-off the corona ions generated by the charger, and consequently the small charged particles are filtered out together with ions used for the charging."

And regarding to P21, L25-26, we added more detailed information: "The corona generated ions below the lowest detection limit should be always cut out of the particle spectra during subsequent data processing. The lowest detection of the NAIS is equal to the upper edge of the corona ion size distribution which is illustrated with the gray shaded area in Fig. 3. The determination of the lower detection limit should be done always when new particle formation (i.e. natural cluster activation and growth) is not taking place. After removing the corona ions from the particle spectra, the particle concentrations in sub-3nm should remain in a concentration range of ~200–700 cm-3."

P5, Section 3: The term "calibration" is used for different purposes with different meanings (e.g. P7,L2 :"the calibration (actually a flow verification)"). This can be misleading.

We agree with the referee's comment, and we remodified the manuscript accordingly. In modifications, we followed the terminology: i) *a calibration* indicates the error of the instrument and compensates for any lack of trueness by applying a correction, and ii) *a verification* indicates that the measurement error is smaller than a so called maximum permissible error.

E.g. we modified the chapter 3.1.1.Determining the flows of the NAIS: Option B, where it was wrong to mention a calibration.

"Option B: Flow calibration with flowmeters for 3- and 4-blower systems" changed to "Option B: Flow verification with flowmeters for 3- and 4-blower systems"

P6, L18-24: Can you give more details regarding the way to adjust the Venturis' pressure drops? This will also help for the 3- and 4- blower systems described on P7 (L21: "The valves should be adjusted accordingly.")

If the balance has drifted, then the pressure drop will be higher than specified on one Venturi, and lower than specified on the other. To fix that, either the valve on the higher pressure drop

side should be slightly closed or the other valve slightly opened. Which valve to adjust is not strictly determined. If the valves are both too open, then small disturbances (e.g. contamination of laminarizing grids) may shift the flow balance too easily. If the valves are both too closed then the sample flow blower may be overstrained.

We added more information on the way to adjust the Venturis' pressure drops (P6, L24): "To adjust the Venturi's pressure drop turn the brass knob which is located on a side of the flow adjusting valve with pliers (middle bottom panel, Fig. 5). To fix misbalanced flows, either the valve on the higher pressure drop side should be slightly closed or the other valve slightly opened."

P7, L14: Can the calibration coefficient be directly adjusted by the user? If yes, can you give additional pieces of explanation? If no, clearly state that assistance from Airel is needed.

**The user can adjust those coefficients but with the assistance from Airel. Thus, we added a sentence: "*To change these calibration coefficients, the user should contact to Airel Ltd.*"**

P10, L10-17: Can you precise the setup which is used for the "drying by heating". Should particular precautions be taken at altitude sites to limit RH in the sample flow? High RH conditions are especially found during in-cloud measurement and can coincide with very low temperatures.

The setup to be used for the "drying by heating" is described already in the manuscript (P10, L12-14). To avoid misunderstandings we modified the sentences and clarified the language: "During earlier deployments in South America (Backman et al. 2012) drying by heating has been used for the NAIS. The inlet line and at the instrument is heated above ambient temperature to avoid water condensing inside the tubing or the instrument. At high altitude sites and other sites with heavy snow storms heated inlet has been used to avoid ice blocking the inlet line. Heating of the sample flow may change the sample by evaporating the most volatile particulate species."

P10, L26-27: Can you give more details regarding the way to take into account the deviation between target and actual flow rates in the processing of the number size distributions?

We added more detailed information which procedures to take to take into account the change in the sample or sheath flow rate. We added sentences: "When the sample-flow rate is recorded, the ion and particle number concentrations can simply be multiplied by (default flow rate)/(recorded or measured flow rate) ratio. When the sheath flow has changed from default values the ion and particle size distributions should be re-inverted (assistance from Airel Ltd. is needed)."

P11, L1-8: In case of measurements performed with a one blower system, are there corrections that can be made on the data afterwards to compensate for an unappropriated use of the blower?

Some correction may be possible if the actual flow rate that the instrument was running with are known. However in the case of the one blower system it is difficult to estimate how the different flowrates behave if the main blower flowrate is wrong or the flow balance is wrong.

This referee comment/question is related to same corrections to number size distributions as replied above (P10, L26-27). Thus, we modified the sentence at P11, L1-8: "*Otherwise, the user needs to keep the blower operating in the right volumetric flow range manually or apply a*

correction to the number size distributions, and a separate airflow calibration is needed for the 1-blower system in the variable environmental conditions."

P12, L21-22: You should precise that the full procedure is described in the supplementary.

We assume this comment refers to P13, L21-22, (a small typo) and to the post-filtering voltage manual adjustment by the user which is described detailed in the supplementary (S2: NAIS troubleshooting by eliminating potential causes of a problem). I added a reference to the Supplement S2: "In the 1-blower systems, the post-filter voltage is adjusted by the user manually, see Supplement S2 for details."

P18, L22-26: I am not sure to understand what the authors mean when talking about "correction". Should we try to correct NAIS data based on SMPS measurements, and if yes how? Or should we just ignore NAIS data above 20 nm, as suggested on P23, L22?

Yes, we recommend to use the particle spectra measured with the NAIS up to 20 nm (P23, L22). To avoid misunderstandings, we modified the sentences as follows (P18, L22-26): "Therefore, we recommend that the ion and particle number size distributions above 20 nm should be utilized with caution. One possibility is to merge the number size distribution measured with the NAIS into a distribution measured with e.g. Differential Mobility Particle Sizer (DMPS, Wiedensohler et al., 2012) in size range from 10 to 1000 nm to obtain information on the background aerosol population to be used in the data inversion (e.g. Kulmala et al., 2012)."

P20, L29: "supplementary data". Be more precised.

We clarified the sentence: "If possible the ion and particle data measured with the NAIS should be crosschecked with the data from additional instruments."

P21, L12: Should corrections be applied even when using the recommended inlet manufactured by Airel?

It is stated earlier in the manuscript (P20, L21-23 and P19, L4-6) that the inversion matrix considers particle losses inside the instrument. Diffusion losses for the sampled particles inside the instrument are estimated by fitting a diffusion length parameter of the instrument model to the ion mode calibration results. The diffusion losses in the sampling lines prior to the instrument should be taken into account by the user. Thus, the users need to take into account only the sampling line losses. Example photo of a sampling line is in Fig. 7. As the NAIS comes from Airel Ltd, without a sampling inlet, we assume that the referee is referring the internal (inlet) tubes considered to be as "inside the instrument". We added more information to manuscript: "*Diffusion losses inside the sampling lines prior, see Fig. 7 (upper panel), to the instrument should be taken into account by post-processing of the data.*"

P21, L25-26: In practice, how can we accurately find out the lowest detectable size given the fact that both residual generated ions and sampled particles are present at the smallest size ranges? This should be clearly explained using Fig. 3 and 4.

**See our earlier reply to comment P5, L1-5.**